# `Pisces`: Cryptography-based Private Retrieval-Augmented Generation with Dual-Path Retrieval

**Xiaojian Liang** [*], **Lushan Song**[*], **Shishuai Du**[*], **Weicheng Zhu, Tan Li Hui Faith, Jun Jie Sim, Haibing Jin, Zhenghao Wu, Yingting Liu, Xin Zhang, Jiang-Ming Yang, Pu Duan** [†]
Ant International, Ant Group

## ABSTRACT

Retrieval-augmented generation (RAG) enhances the response quality of large language models (LLMs) when handling domain-specific tasks, yet raises significant privacy concerns. This is because both the user query and documents within the knowledge base often contain sensitive or confidential information. To address these concerns, we propose `Pisces`, the first practical cryptography-based RAG framework that supports dual-path retrieval, while protecting both the query and documents. Along the semantic retrieval path, we reduce computation and communication overhead by leveraging a coarse-to-fine strategy. Specifically, a novel oblivious filter is used to privately select a candidate set of documents to reduce the scale of subsequent cosine similarity computations. For the lexical retrieval path, to reduce the overhead of repeatedly invoking labeled PSI, we implement a multi-instance labeled PSI protocol to compute term frequencies for BM25 scoring in a single execution. `Pisces` can also be integrated with existing privacy-preserving LLM inference frameworks to achieve end-to-end privacy. Experiments demonstrate that `Pisces` achieves retrieval accuracy comparable to the plaintext baselines, within a 1.87% margin. Our code is available on GitHub[1].

## 1 INTRODUCTION

Although large language models (LLMs) (Achiam et al., 2023; Liu et al., 2024) have achieved remarkable success in natural language processing tasks, they still exhibit significant limitations in domain-specific tasks, for example, healthcare diagnostics. In particular, LLMs may produce hallucinations due to a lack of domain-specific knowledge (Huang et al., 2025; Li et al., 2024a). To mitigate these limitations, retrieval-augmented generation (RAG) (Lewis et al., 2020; Gao et al., 2023; Jiang et al., 2023) has emerged as a promising paradigm. RAG mainly consists of two processes: retrieval and generation. Specifically, it first retrieves relevant documents from external knowledge bases and then generates a higher-quality response by integrating the query with the retrieved documents.

However, RAG systems, which involve private data, raise significant privacy concerns (Huang et al., 2023; Zeng et al., 2024). For instance, in personalized healthcare diagnostics, a healthcare agent with an LLM as the cornerstone for which personalized queries regarding one's health are made against, supported by an RAG system that encompasses a medical knowledge base. The user interacting with the agent would be highly concerned about revealing too much personal information, such as family history, for fear of any potential privacy exposure. Simultaneously, queries could reveal information about the individuals part of the RAG knowledge base, which contains highly sensitive personal information that is hard to anonymize in nature, such as rare diseases, clinical notes, or biometric identifiers. These works (Zeng et al., 2024; Huang et al., 2023) demonstrate that it is possible to extract sentences verbatim or personally identifiable information from the knowledge base. The leakage of any sensitive information would violate data privacy regulations, such as GDPR (Voigt & Von dem Bussche, 2017), PIPL (Congress, a), and HIPAA (Congress, b). This highlights the need for privacy in a reciprocal manner, where, during retrieval, the knowledge base does not learn additional information about the user, and the user does not learn additional information about the knowledge base.

---

[*]Equal contribution.

[†]Correspondence to p.duan@ant-intl.com

[1]`https://github.com/liang-xiaojian/Pisces`

Table 1: Comparison with prior works.

| Framework | Retrieval Path | | Privacy | | Mechanism |
|---|---|---|---|---|---|
| | Semantic | Lexical | Query | Documents | |
| DP-RAG (Grislain, 2025) | ✓ | ✗ | ✗ | ✓ | DP |
| RemoteRAG (Cheng et al., 2024) | ✓ | ✗ | ✓ | ✗ | DP, Cryptography |
| (Yao & Li, 2025) | ✓ | ✗ | ✓ | ✓ | DP |
| Pisces (Ours) | ✓ | ✓ | ✓ | ✓ | Cryptography |

While recent substantial works (Rovida & Leporati, 2024; Moon et al., 2024; Xu et al., 2025; Lu et al., 2023) have focused on private generation in RAG systems, the private retrieval process remains comparatively underexplored. Besides, as summarized in Table 1, existing works on the private retrieval process (Grislain, 2025; Cheng et al., 2025; Yao & Li, 2025) primarily apply differential privacy (DP). However, these works typically exhibit limited retrieval performance due to support for only semantic retrieval and struggle with lexical retrieval (e.g., BM25). This is because the noise introduced by DP inherently disrupts exact term matching. Works like (Kuzi et al., 2020) suggest that dual-path retrieval, a combination of semantic and lexical retrieval, achieves better retrieval performance.

Consequently, we aim to tackle a challenging question: *How can we support dual-path retrieval while ensuring privacy for both the query and documents during the retrieval process?*

We consider a cryptography-based solution to address the question above. Firstly, cryptography-based technologies, such as secure multi-party computation (MPC), support both the similarity computations required by semantic retrieval and the exact term matching essential for lexical retrieval, addressing the key limitations of DP-based approaches. This is because cryptography-based technologies do not alter the raw data, while DP involves perturbing the data with irreversible noise, which makes it difficult to perform exact matching. Secondly, during the entire retrieval process, cryptography-based techniques ensure that any raw data exchanged is in an encrypted form, protecting the privacy of both the query and documents.

Unfortunately, existing cryptography-based technologies suffer from two challenges in terms of efficiency when directly deployed to the dual-path retrieval. (1) For the semantic retrieval, direct computation of similarities between a query and all documents in a large-scale knowledge base incurs prohibitive computation and communication overhead. (2) For the lexical retrieval, state-of-the-art labeled private set intersection (PSI) methods require multiple invocations to obtain all necessary term frequencies for BM25 scoring, leading to significant computational overhead.

To address these two challenges, we propose a cryptography-based framework, Pisces, that introduces two customized cryptography-based protocols for significant efficiency improvements. (1) For the semantic retrieval, we introduce a novel oblivious filter protocol as the first step of our adopted coarse-to-fine strategy. This protocol privately selects a candidate set to substantially reduce the search space. Then, we conduct private cosine similarity computations between the query and candidates, utilizing MPC primitives. (2) For the lexical retrieval path, we design a multi-instance labeled PSI protocol that obtains all necessary term frequencies in a single execution, avoiding the overhead of repeated labeled PSI invocations. Pisces provides strong privacy guarantees for both the query and documents while maintaining high retrieval performance, offering a practical solution for privacy-sensitive RAG applications.

Our contributions are summarized as follows:

- To the best of our knowledge, we are the first to propose the cryptography-based RAG retrieval framework, Pisces, with dual-path retrieval, while ensuring privacy for both the query and documents.

- We propose two customized cryptography-based protocols for significant efficiency improvements. (1) We adopt a coarse-to-fine strategy for the semantic retrieval path with a novel oblivious filter to reduce computation and communication overhead. (2) We design an efficient multi-instance labeled PSI protocol to avoid the cost of repeated labeled PSI invocations.

We conducted comprehensive experiments to evaluate the performance of `Pisces`. For accuracy, the results show that `Pisces` achieves retrieval accuracy comparable to plaintext baselines over the ground-truth of the dataset, within a 1.87% margin. At the same time, we observe that combining semantic and lexical paths significantly improves retrieval accuracy. For efficiency, the experiments demonstrate that our coarse-to-fine strategy saves retrieval time by 41.21%, reduces upload and download overhead by 68.77% compared to the fine-only strategy on the large-scale dataset. Additionally, our proposed multi-instance labeled PSI outperforms state-of-the-art labeled PSI protocol (Yang et al., 2024), achieving $496.03\times$ speedup in runtime, and reducing upload and download overhead by $70733\times$ and $2.84\times$, respectively. Overall, `Pisces` is practical in both accuracy and efficiency.

It's worth noting that while `Pisces` focuses on privacy preservation during the retrieval process, it can seamlessly integrate with existing privacy-preserving LLM inference frameworks to achieve end-to-end private preservation in the whole RAG system.

## 2 PRELIMINARIES

In this work, we use a variety of cryptographic primitives to achieve a private RAG retrieval process. Below we briefly summarize each primitive, and further details can be found in Appendix A.1.

- **Secure Multi-Party Computation** (Ma et al., 2023). A cryptographic technology that enables multiple mutually distrustful parties to cooperatively compute a predefined function while keeping their data private.
- **Secret Sharing** (Keller, 2020). A critical primitive of MPC, that breaks a secret value into multiple shares held by different parties. The secret value can only be reconstructed when a sufficient number of shares are combined.
- **Labeled Private Set Intersection** (Chen et al., 2018). PSI (Jarecki & Liu, 2010) allows two parties to learn the intersection of their sets without revealing any information outside the intersection. Labeled PSI extends the traditional PSI by returning the label that is associated with each element in the intersection.
- **Oblivious Pseudorandom Function (OPRF)** (Naor et al., 1999). Enables two parties to jointly compute a pseudorandom function such that one party learns the output, while the other learns nothing about the input or output.
- **Oblivious Key-Value Store (OKVS)** (Garimella et al., 2021). A data structure that encodes a set of key-value pairs into a compact representation while preserving the privacy of both keys and values.
- **Batch private information retrieval-to-Share (PIR-to-share)** (Song et al., 2025). A cryptographic primitive that enables a client to privately retrieve the values corresponding to its queries from the server. After the execution, both parties obtain the secret shares of the retrieved values.

Additionally, we provide detailed descriptions of the semantic similarity (Awasthy et al., 2025; Zhang et al., 2025) and BM25 (Robertson et al., 2009; Lù, 2024) for lexical retrieval in Appendix A.2 and Appendix A.3, respectively.

## 3 PROBLEM DEFINITION AND THREAT MODEL

We formally define the problem of private retrieval in RAG systems, followed by the threat model considered in this paper. Additionally, the Table 6 in Appendix B summarizes the frequently used notation in this paper.

### 3.1 PROBLEM DEFINITION

We consider a setting with two parties: a server $\mathcal{S}$ and a user $\mathcal{C}$. The server $\mathcal{S}$ holds a knowledge base that contains a large corpus of sensitive textual documents $D$. The user $\mathcal{C}$ submits a private query $Q$ to $\mathcal{S}$. Let $\mathcal{R}$ denote the retrieval module and $D^K$ denote the set of top-$K$ documents retrieved from $D$. Then the retrieval process in our paper is defined as: $Enc(D^K) = \mathcal{R}(Q, D)$. During the retrieval process, `Pisces` ensures that neither party learns the other's sensitive information.

## 3.2 THREAT MODEL

We consider a semi-honest adversary, where the two parties $\mathcal{S}$ and $\mathcal{C}$ follow the protocol honestly, but are curious about information the other party is holding. Our threat model focuses on the private information leakage during the retrieval process in the RAG system. We protect the privacy of both the user query and the knowledge base of the server.

*Protection of the user query.* During the retrieval process, the server cannot directly access the user query. Moreover, the server remains unaware of which specific documents are retrieved, thereby preventing any inference of sensitive user information based on retrieval results.

*Protection of the knowledge base.* During the retrieval process, the user cannot obtain sensitive information within the documents in the knowledge base, nor know which documents were retrieved.

## 4 PROPOSED METHOD

### 4.1 OVERVIEW

`Pisces` involves two parties: a server $\mathcal{S}$, who holds a sensitive knowledge base (a large corpus of textual documents $D$), and a user $\mathcal{C}$, who holds a private query $Q$.

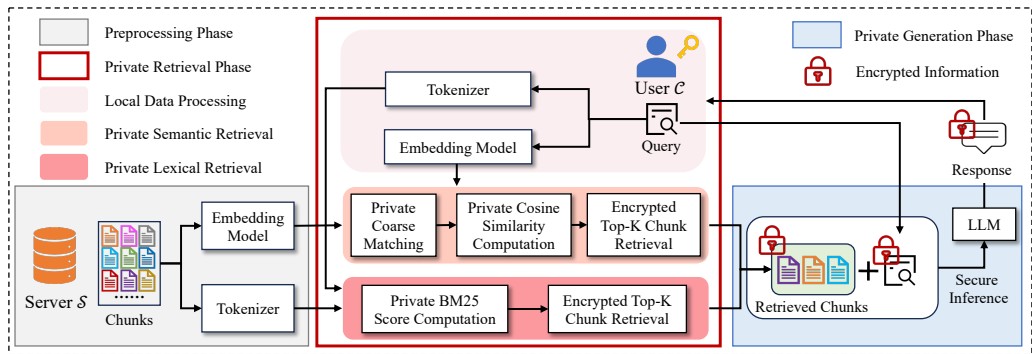

Figure 1: Overview of our proposed `Pisces`, where the private retrieval phase is our core contribution.

As shown in Figure 1, the whole process of `Pisces` consists of three phases:

**Phase 1: Preprocessing Phase.** In this phase, $\mathcal{S}$ preprocesses its private document corpus $D$ for efficient retrieval.

- **Document Chunking.** $\mathcal{S}$ break down $D$ into $N$ smaller chunks of text, i.e. $D = \{c_1, c_2, \ldots, c_N\}$.
- **Vector Embedding.** $\mathcal{S}$ encodes each chunk $c_i$ ($i \in [1,N]$) into vector representations using an embedding model, resulting in $D^v = \{I_i; \mathbf{v}_i; c_i\}_{i \in [1,N]}$, where $I_i$ and $\mathbf{v}_i$ are the index and vector representation corresponding to the chunk $c_i$, respectively.
- **Tokenization & Term Frequencies.** $\mathcal{S}$ tokenizes each chunk $c_i$ ($i \in [1,N]$) with a tokenizer and computes the term frequencies, resulting in $D^t = \left\{I_i; \{w_{i,l} : tf_{i,l}\}_{l \in [1,m_i]}; c_i\right\}_{i \in [1,N]}$, where $m_i$ is the total number of unique tokens of $c_i$, $w_{i,l}$ is the $l$-th token in the chunk $c_i$ and $tf_{i,l}$ is its term frequency.

**Phase 2: Private Retrieval Phase.** In this phase, $\mathcal{S}$ interacts with $\mathcal{C}$ to retrieve the relevant chunks for the query $Q$ with privacy preservation.

- **Local Data Processing.** $\mathcal{C}$ locally encodes its query $Q$ into vector representations $\mathbf{q}$ and tokenizes $Q$ to $n$ tokens, i.e. $Q^t = \{q_1, q_2, \ldots, q_n\}$, utilizing the same embedding model and tokenizer applied during the preprocessing phase.

- **Private Semantic Retrieval.** $\mathcal{S}$ and $\mathcal{C}$ invoke the private semantic similarity protocol $\prod_{\text{PrivateSS}}$ (Protocol 1), where $\mathcal{S}$ inputs $D^v$ and $\mathcal{C}$ inputs $\mathbf{q}$. After execution, $\mathcal{S}$ obtains the encrypted top-$K$ chunks with the highest similarity scores.
- **Private Lexical Retrieval.** $\mathcal{S}$ and $\mathcal{C}$ invoke the private BM25 protocol $\prod_{\text{PrivateBM25}}$ (Protocol 2), where $\mathcal{S}$ inputs $D^t$ and $\mathcal{C}$ inputs $Q^t$. After execution, $\mathcal{S}$ obtains the encrypted top-$K$ chunks with the highest BM25 scores.

**Phase 3: Private Generation Phase.** In this phase, $\mathcal{C}$ obtains the response to its query $Q$ while preserving privacy.

- **Context Fusion.** Then $\mathcal{S}$ fuses the encrypted retrieved $2K$ chunks with the encrypted query.
- **Secure Inference.** $\mathcal{S}$ and $\mathcal{C}$ execute the secure LLM inference framework to generate an encrypted response to $\mathcal{C}$.

Notably, in this paper, we pay attention to the preprocessing phase and the private retrieval phase, where the private retrieval phase is our core contribution. Furthermore, `Pisces` can be integrated with the existing secure inference framework based on various technologies, such as HE (Rovida & Leporati, 2024; Moon et al., 2024), MPC (Xu et al., 2025; Lu et al., 2023; Pang et al., 2024), and DP (Koga et al., 2024), to achieve end-to-end privacy.

## 4.2 PRIVATE SEMANTIC RETRIEVAL

Semantic retrieval aims to retrieve the top-$K$ most semantically relevant chunks for a query issued by a user $\mathcal{C}$ from a set of chunks held by a server $\mathcal{S}$. We design an efficient and private semantic similarity protocol $\prod_{\text{PrivateSS}}$ (Protocol 1) that leverages a coarse-to-fine pipeline. Direct computation of the cosine similarity over the entire set of chunks with cryptographic protocols (e.g., MPC) is prohibitively expensive. To mitigate this, we first propose a novel oblivious filter (Protocol 3, described in Appendix C) that privately selects a subset of candidate chunks, significantly reducing the scale of subsequent cosine similarity computations.

---

**Protocol 1: $\prod_{\text{PrivateSS}}$**

**Input:** $\mathcal{S}$ inputs the embedded chunk set $D^v = \{I_i; \mathbf{v}_i; c_i\}_{i \in [1,N]}$, where $I_i$ and $\mathbf{v}_i$ are the index and vector representation corresponding to the chunk $c_i$, respectively. $\mathcal{C}$ inputs embedded query $\mathbf{q}$.

**Output:** $\mathcal{S}$ learns the encrypted top-$K$ chunks $\{Enc(c_{t1}), Enc(c_{t2}), \ldots, Enc(c_{tK})\}$ with the highest cosine similarities.

1: $\mathcal{S}$ computes $\mathbf{v}_i^b \leftarrow SimHash(\mathbf{v}_i) = \{0,1\}^{\mathcal{L}}$ for $i \in [1,N]$. $\mathcal{C}$ computes $\mathbf{q}^b \leftarrow SimHash(\mathbf{q}) = \{0,1\}^{\mathcal{L}}$.

2: $\mathcal{S}$ and $\mathcal{C}$ invoke the obvious filter $\prod_{\text{Oblivious\_Filter}}$ (Protocol 3) with $\{I_i; \mathbf{v}_i^b; c_i\}_{i \in [1,N]}$ and $\mathbf{q}^b$ as input, respectively. After execution, $\mathcal{S}$ learns the candidate chunk set $D'$.

3: $\mathcal{S}$ and $\mathcal{C}$ securely compute the cosine similarity between each chunk in $D'$ and the query using MPC protocols based on secret sharing (Ma et al., 2023), obtaining secret shares of the cosine similarities, respectively.

4: $\mathcal{S}$ and $\mathcal{C}$ invoke the secure sorting protocol (Li et al., 2024b) with the secret shares of cosine similarities as input. After execution, $\mathcal{C}$ learns the indices $I^K = \{I_{t1}, I_{t2}, \ldots, I_{tK}\}$ of top-$K$ chunks with the highest cosine similarities.

5: $\mathcal{S}$ and $\mathcal{C}$ invoke the batch PIR-to-share protocol (Song et al., 2025) with $D^v$ and $I^K$ as input, respectively. After execution, $\mathcal{S}$ and $\mathcal{C}$ learn the secret shares $\langle D^K \rangle$ of top-$K$ chunks corresponding to $I^t$, where $D^K = \{c_{t1}, c_{t2}, \ldots, c_{tK}\}$.

6: $\mathcal{C}$ encrypts $\langle D^K \rangle^C$ to $Enc\left(\langle D^K \rangle^C\right)$ using FHE and sends it to $\mathcal{S}$. $\mathcal{S}$ computes $Enc(D^K) \leftarrow Enc\left(\langle D^K \rangle^C\right) + \langle D^K \rangle^S$.

---

We describe the private semantic similarity protocol $\prod_{\text{PrivateSS}}$ (Protocol 1) as follows:

- **Step 1 (Lines 1-2) Private Coarse Matching.** To leverage the computational efficiency of Hamming distance in cryptographic protocols, particularly for large-scale knowledge bases, we first

translate cosine similarity computations into Hamming distance. Concretely, $\mathcal{S}$ and $\mathcal{C}$ convert their vector embeddings $\mathbf{v}_i$ ($i \in [1, N]$) and $\mathbf{q}$ into $\mathcal{L}$-bit binary vectors $\mathbf{v}_i^b$ and $\mathbf{q}^b$, respectively, using SimHash (Charikar, 2002). They then invoke the obvious filter $\prod_{\text{Oblivious\_Filter}}$ (Protocol 3) that operates over Hamming space to identify a candidate set of chunks, which is much smaller than the full chunk set, without revealing any sensitive information about the query or knowledge base.

- **Step 2 (Line 3) Private Cosine Similarity Computation.** After identifying the candidate set of chunks, $\mathcal{S}$ and $\mathcal{C}$ perform fine-grained matching by jointly computing the cosine similarity between each candidate chunk and the query utilizing MPC protocols (Ma et al., 2023) based on secret sharing.

- **Step 3 (Lines 4-6) Encrypted Top-$K$ Chunk Retrieval.** Given the computed cosine similarities, $\mathcal{S}$ and $\mathcal{C}$ privately retrieve the corresponding top-$K$ encrypted chunks. Concretely, $\mathcal{C}$ first obtains the indices of the top-$K$ chunks with the highest cosine similarities utilizing a secure sorting protocol (Li et al., 2024b). $\mathcal{S}$ and $\mathcal{C}$ then retrieve these chunks in secret-shared form utilizing a batch PIR-to-share protocol (Song et al., 2025). Finally, they convert the secret shares of top-$K$ chunks into homomorphic encryption ciphertexts. This conversion is optional and depends on the input type of the subsequent secure LLM inference framework.

## 4.3 PRIVATE LEXICAL RETRIEVAL

Lexical matching adopted in this paper considers an alternative scoring metric as described in A.3 for the top-$K$ chunks. To achieve lexical matching efficiently and privately, we design an efficient private BM25 protocol. We first explore labeled PSI to privately obtain term frequencies for BM25 scoring. Furthermore, to reduce the overhead of repeatedly invoking labeled PSI for each chunk, we introduce a multi-instance labeled PSI protocol $\prod_{\text{MultLPSI}}$ (Protocol 4, and the details are shown in Appendix D) based on OPRF and OKVS, that computes all per-chunk query term frequencies in a single execution.

We describe the private BM25 protocol $\prod_{\text{PrivateBM25}}$ (Protocol 2) as follows:

- **Step 1 (Lines 1-4) Private BM25 Scores Computation.** Firstly, $\mathcal{C}$ privately obtains the term frequency of each query token in each chunk by invoking the multi-instance labeled PSI protocol (Protocol 4). From these term frequencies, $\mathcal{C}$ could compute the document frequency (i.e., the number of chunks in which $q_j$ appears) for each query token $q_j$. Then $\mathcal{S}$ and $\mathcal{C}$ jointly compute the BM25 score for each chunk utilizing MPC protocols based on secret sharing (Ma et al., 2023).

- **Step 2 (Lines 5-7) Encrypted Top-$K$ Chunk Retrieval.** Given the computed BM25 scores, $\mathcal{S}$ and $\mathcal{C}$ privately retrieve the corresponding top-$K$ encrypted chunks. This step is similar to Step 3 in the private similarity matching protocol $\prod_{\text{PrivateSS}}$ (Protocol 1) and therefore we omit the details here.

## 4.4 PRIVATE GENERATION

`Pisces` can be integrated with various secure LLM inference frameworks.

**Integrate with HE-based Secure Inference Frameworks.** As discussed in Section 3.1, $\mathcal{S}$ receives the homomorphically encrypted retrieved chunks along with the encrypted query. It then executes the HE-based secure LLM inference framework Rovida & Leporati (2024); Moon et al. (2024) to compute an encrypted response, which is subsequently returned to $\mathcal{C}$.

**Integrate with MPC-based Secure Inference Frameworks.** $\mathcal{S}$ and $\mathcal{C}$ avoid converting the secret shares of the retrieved chunks into homomorphic ciphertexts, skipping Step 6 of the private semantic similarity protocol (Protocol 1) and Step 7 of the private BM25 protocol (Protocol 2). Instead, $\mathcal{C}$ secret shares its query with $\mathcal{S}$. They then use these shares directly to execute the MPC-based secure LLM inference framework (Xu et al., 2025; Lu et al., 2023; Pang et al., 2024), thereby jointly computing secret shares of the response.

**Integrate with DP-based Secure Inference Frameworks.** Upon receiving both the homomorphically encrypted retrieved chunks and the encrypted query, $\mathcal{S}$ injects differential privacy noise into the received encrypted result. This perturbed result is then sent to $\mathcal{C}$, who decrypts it and proceeds with the DP-based secure LLM inference framework (Koga et al., 2024) to produce the response.

---

**Protocol 2:** $\prod_{\text{PrivateBM25}}$

**Input:** $\mathcal{S}$ inputs the tokenized chunk set $D^t = \left\{ I_i; \{w_{i,l} : tf_{i,l}\}_{l \in [1,m_i]}; c_i \right\}_{i \in [1,N]}$, where $m_i$ is the total number of unique tokens of $c_i$, $w_{i,l}$ is the $l$-th token of chunk $c_i$ and $tf_{i,l}^D$ is its term frequency. $\mathcal{C}$ inputs tokenized query $Q^t = \{q_1, q_2, \dots, q_n\}$, where $n$ is the number of tokens in $Q$.

**Output:** $\mathcal{S}$ learns the encrypted top-$K$ chunks $\{Enc(c_{t1}), Enc(c_{t2}), \dots, Enc(c_{tK})\}$ with the highest BM25 scores.

1: $\mathcal{S}$ and $\mathcal{C}$ invoke the multi-instance labeled PSI protocol $\prod_{\text{MultLPSI}}$ (Protocol 4) with $\{w_{i,l} : tf_{i,l}\}_{i \in [1,N], l \in [1,m_i]}$ and $Q^t$ as input, respectively. After execution, $\mathcal{C}$ learns the term frequency $tf'_{i,j}$ of each token $q_j$ ($j \in [1,n]$) in each chunk $c_i$ ($i \in [1,N]$), where if $q_j = w_{i,l}$, $tf'_{i,j} \leftarrow tf_{i,l}$, and otherwise $tf'_{i,j} = 0$.

2: $\mathcal{C}$ computes the document frequency $df_j \leftarrow \sum_{i=1}^{N}(tf'_{i,j} > 0?1:0)$ for each token $q_j$ ($j \in [1,n]$).

3: $\mathcal{C}$ and $\mathcal{S}$ locally computes $\log\left(1 + \frac{N - df_j + 0.5}{df_j + 0.5}\right) \cdot tf'_{i,j}$ and $k_1 \cdot \left(1 - b + b \cdot \frac{L_{c_i}}{L_{ave}}\right)$, respectively, for $i \in [1,N]$ and $j \in [1,n]$.

4: $\mathcal{S}$ and $\mathcal{C}$ secure computes the BM25 scores according Equation (1) utilizing MPC protocols based on secret sharing (Ma et al., 2023). Then $\mathcal{S}$ and $\mathcal{C}$ learns the secret shares of BM25 scores, respectively.

5: $\mathcal{S}$ and $\mathcal{C}$ invoke the secure sorting protocol (Li et al., 2024b) with the secret shares of BM25 scores as input. After execution, $\mathcal{C}$ learns the indices $I^K = \{I_{t1}, I_{t2}, \dots, I_{tK}\}$ of tok-$K$ chunks with the highest BM25 scores.

6: $\mathcal{S}$ and $\mathcal{C}$ invoke the batch PIR-to-share protocol (Song et al., 2025) with $D^v$ and $I^K$ as input, respectively. After execution, $\mathcal{S}$ and $\mathcal{C}$ learn the secret shares $\langle D^K \rangle$ of top-$K$ chunks corresponding t $I^t$, where $D^K = \{c_{t1}, c_{t2}, \dots, c_{tK}\}$.

7: $\mathcal{C}$ encrypts $\langle D^K \rangle^C$ to $Enc\left(\langle D^K \rangle^C\right)$ using FHE and sends it to $\mathcal{S}$. $\mathcal{S}$ computes $Enc(D^K) \leftarrow Enc\left(\langle D^K \rangle^C\right) + \langle D^K \rangle^S$.

---

## 5 EXPERIMENTS

In this section, we first introduce the experimental settings. Then we evaluate the practicality of `Pisces` in two parts: (1) the accuracy of `Pisces` compared to the plaintext baseline, and (2) the efficiency of `Pisces` compared to state-of-the-art cryptographic techniques.

### 5.1 EXPERIMENTAL SETTINGS

**Embedding Model and Tokenizer.** We employ an open-source embedding model, granite-embedding-small-english-r2[2] (Awasthy et al., 2025) to encode chunks and the query into 384-dimensional vector representations. Additionally, we utilize an open-source tokenizer BERT[3] (Devlin et al., 2019) for chunk and query tokenization.

**Datasets.** We use three datasets: ClapNQ, SQuAD, and HotpotQA as RAG datasets. The details of these datasets are shown in Table 7. For the Dev_answerable dataset (300 queries in total), we run 300 queries and take the average to obtain stable results, while for the other datasets, we run 1,000 queries.

**Baselines.** To demonstrate the accuracy of `Pisces`, we compare `Pisces` against the plaintext baseline and DP-based approaches listed in Table 1 under the same RAG architecture. To demonstrate efficiency, we compare the semantic retrieval of `Pisces` against a semantic retrieval baseline without coarse matching and the lexical retrieval of `Pisces` against a lexical retrieval baseline with the labeled PSI protocol LSE (Yang et al., 2024).

---

[2] https://huggingface.co/ibm-granite/granite-embedding-small-english-r2
[3] https://github.com/google-research/bert

**Environment.** All of our experiments are conducted on an Apple M4 Pro machine with 24 GB of RAM, running macOS 15.6.1 (24G90).

## 5.2 ACCURACY EVALUATION

We evaluate the accuracy of `Pisces` against the plaintext baseline through two complementary approaches.

First, for each of the two retrieval paths, we compare the chunks retrieved by `Pisces` with those by the corresponding plaintext retrieval paths. Tables 2 and 3 present semantic and lexical retrieval accuracy under the Top-5 and Top-10 settings, respectively, compared to the plaintext baseline. The results demonstrate that `Pisces` achieves semantic retrieval accuracy ranges from 78.02% to 90.44% for Top-5, and from 74.86% to 86.83% for Top-10. At the same time, lexical retrieval accuracy ranges from 85.72% to 98.22 % for Top-5 and from 86.44% to 98.02% for Top-10. The accuracy drop in the semantic retrieval path mainly stems from the information loss when approximating cosine similarity with Hamming distance via SimHash. The slight degradation in lexical retrieval accuracy is primarily due to precision loss during secure BM25 score computation.

Table 2: Semantic retrieval accuracy against the plaintext baseline.

| Dataset | | Top-5 | | Top-10 | |
|---|---|---|---|---|---|
| | | Accuracy | Time (s) | Accuracy | Time (s) |
| ClapNQ | Dev_answerable | 87.67% | 3.47 | 86.83% | 3.56 |
| | Train_answerable | 80.30% | 4.12 | 77.78% | 4.17 |
| | Train_single_answerable | 90.44% | 7.33 | 81.39% | 7.88 |
| SQuAD | Dev_v2.0 | 78.02% | 3.37 | 75.96% | 3.41 |
| | Training_v2.0 | 78.14% | 4.38 | 74.86% | 4.46 |
| HotpotQA | Dev_distractor | 79.90% | 18.91 | 79.80% | 20.10 |
| | Dev_fullwiki | 79.46% | 19.27 | 78.23% | 20.76 |
| | Training | 82.92% | 147.08 | 81.42% | 160.90 |

Table 3: Lexical retrieval accuracy against the plaintext baseline.

| Dataset | | Top-5 | | Top-10 | |
|---|---|---|---|---|---|
| | | Accuracy | Time (s) | Accuracy | Time (s) |
| ClapNQ | Dev_answerable | 97.47% | 1.40 | 96.53% | 1.44 |
| | Train_answerable | 95.62% | 2.07 | 95.13% | 2.24 |
| | Train_single_answerable | 95.64% | 5.94 | 94.99% | 6.86 |
| SQuAD | Dev_v2.0 | 97.56% | 1.39 | 97.32% | 1.42 |
| | Training_v2.0 | 98.22% | 2.59 | 98.02% | 2.82 |
| HotpotQA | Dev_distractor | 90.06% | 21.46 | 89.48% | 25.02 |
| | Dev_fullwiki | 90.58% | 21.39 | 89.85% | 25.61 |
| | Training | 85.72% | 238.60 | 86.44% | 265.91 |

Second, we evaluate the chunks retrieved by both `Pisces` and the plaintext baseline against the dataset ground-truth. Figure2 and Figure5 (Appendix F) compare the top-5 and top-10 retrieval accuracy between `Pisces` and the plaintext baseline, respectively. The results demonstrate that (1) `Pisces` achieves retrieval accuracy comparable to that of the plaintext baseline, and (2) combining semantic and lexical retrieval improves overall retrieval performance. Furthermore, we evaluate the retrieval accuracy of our proposed `Pisces` against existing DP-based approaches listed in Table 1. The detailed top-5 and top-10 accuracy results, presented in Table 8 and Table 9 respectively, demonstrate that `Pisces` achieves significantly superior retrieval accuracy compared to DP-based approaches.

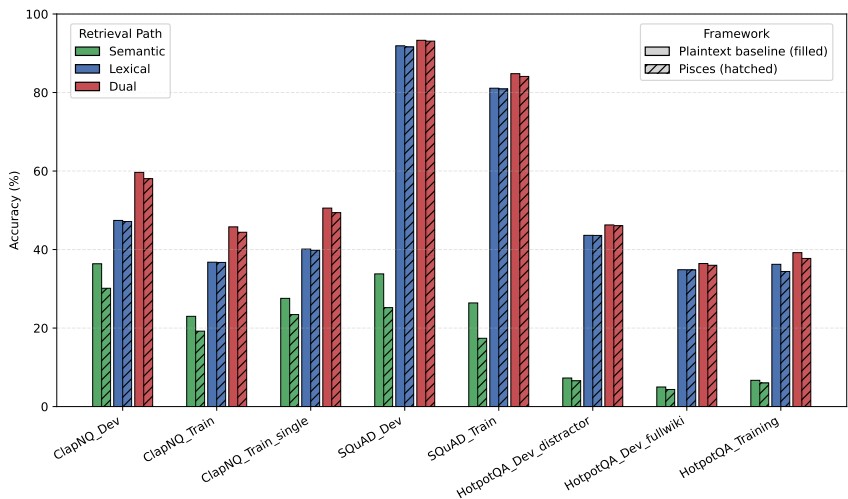

Figure 2: Top-5 Retrieval accuracy comparisons between `Pisces` and plaintext baseline over ground-truth.

## 5.3 EFFICIENCY EVALUATION

We evaluate the efficiency of the two retrieval paths of `Pisces`, respectively.

Table 4: Efficiency comparisons with Fine-only Strategy

| Dataset | | **Fine-Only Strategy** | | | |
| | | Time (s) | Upload (MB) | Download (MB) | Accuracy |
| --- | --- | --- | --- | --- | --- |
| | Dev_answerable | 1.855 | 26.64 | 36.41 | 100.00% |
| ClapNQ | Train_answerable | 2.82 | 69.47 | 230.82 | 99.83% |
| | Train_single_answerable | 10.29 | 278.77 | 1195.62 | 99.94% |
| SQuAD | Dev_v2.0 | 1.714 | 24.06 | 24.147 | 99.64% |
| | Train_v2.0 | 3.66 | 87.97 | 315.18 | 99.95% |
| HotpotQA | Dev_distractor | 34.19 | 1008.39 | 4610.26 | 99.94% |
| | Dev_fullwiki | 33.91 | 1031.45 | 4719.76 | 99.97% |
| **Dataset** | | **Coarse-to-Fine Strategy** | | | |
| | | Time (s) | Upload (MB) | Download (MB) | Accuracy |
| | Dev_answerable | 3.56 | 23.95 | 23.32 | 86.83% |
| ClapNQ | Train_answerable | 4.17 | 36.35 | 85.64 | 77.78% |
| | Train_single_answerable | 7.88 | 113.07 | 442.97 | 81.39% |
| SQuAD | Dev_v2.0 | 3.41 | 21.96 | 13.93 | 75.96% |
| | Train_v2.0 | 4.46 | 43.51 | 118.29 | 74.86% |
| HotpotQA | Dev_distractor | 20.10 | 324.90 | 1439.70 | 79.80% |
| | Dev_fullwiki | 20.76 | 330.69 | 1467.45 | 78.23% |

For the semantic retrieval path, we evaluate the efficiency of our proposed coarse-to-fine strategy against a fine-only baseline (without coarse matching), with both implemented using encrypted computations. The results shown in Table 4 demonstrate that for a large-scale dataset, the coarse-to-fine strategy significantly saves retrieval time by $38.78\% \sim 41.21\%$, reduces the upload and download overhead by $67.53\% \sim 67.78\%$ and $68.49\% \sim 68.77\%$, respectively. In contrast, on small-scale datasets, the fine-only strategy outperforms ours, as the coarse matching step itself, rather than cosine similarity computation, becomes the computational bottleneck.

For the lexical retrieval path, we evaluate the efficiency of our proposed multi-instance labeled PSI protocol $\prod_{\text{MultLPSI}}$ (Protocol 4) with the state-of-the-art labeled PSI protocol LSE (Yang et al.,

Table 5: Efficiency comparisons with labeled PSI

| Dataset | | Labeled PSI | | |
| --- | --- | --- | --- | --- |
| | | Time (s) | Upload (MB) | Download (MB) |
| ClapNQ | Dev_answerable | 3.89 | 1.62 | 0.60 |
| | Train_answerable | 27.57 | 4.21 | 11.35 |
| | Train_single_answerable | 138.89 | 21.22 | 57.77 |
| SQuAD | Dev_v2.0 | 3.15 | 0.48 | 2.03 |
| | Train_v2.0 | 45.89 | 6.99 | 30.05 |
| HotpotQA | Dev_distractor | 1051.98 | 161.61 | 382.26 |
| | Dev_fullwiki | 1179.58 | 176.89 | 414.16 |
| Dataset | | Multi-instance Labeled PSI | | |
| | | Time (s) | Upload (MB) | Download (MB) |
| ClapNQ | Dev_answerable | 0.009 | 0.0003 | 0.58 |
| | Train_answerable | 0.056 | 0.0003 | 4.00 |
| | Train_single_answerable | 0.28 | 0.0003 | 21.04 |
| SQuAD | Dev_v2.0 | 0.008 | 0.0004 | 1.49 |
| | Train_v2.0 | 0.099 | 0.0004 | 22.64 |
| HotpotQA | Dev_distractor | 2.35 | 0.0006 | 79.82 |
| | Dev_fullwiki | 2.59 | 0.0006 | 81.44 |

2024). The results shown in Table 4 demonstrate that our proposed multi-instance labeled PSI outperforms LSE by up to $496.03\times$, $70733\times$, and $2.84\times$ in running time, upload overhead, and download overhead, respectively.

## 6 RELATED WORK

**RAG with Dual-Path Retrieval.** Multiple works (Kuzi et al., 2020; Gao et al., 2021; Li et al., 2022) demonstrate that leveraging semantic and lexical retrieval together significantly improves retrieval performance. Inspired by this, we aim to design `Pisces` that privately supports dual-path retrieval to guarantee the retrieval accuracy.

**RAG with Retrieval Process Protection.** Recent work applies differential privacy by injecting noise into embeddings to protect privacy during the retrieval process. Several works (Grislain, 2025; He et al., 2025) focus on protecting documents during the semantic retrieval, while Cheng et al. (Cheng et al., 2025) propose RemoteRAG to protect the query. Yao and Li (Yao & Li, 2025) further attempt to protect both the query and documents. However, all of these works only consider a single retrieval path, i.e., semantic retrieval. In contrast, `Pisces` supports dual-path retrieval, semantic and lexical, while protecting both the query and documents.

## 7 CONCLUSION

In this paper, we propose `Pisces`, the first practical cryptography-based RAG framework that supports dual-path retrieval while protecting both the query and documents. We design novel cryptographic protocols tailored for efficient semantic and lexical retrieval: a coarse-to-fine semantic strategy that employs a novel oblivious filter over Hamming distance, and an efficient multi-instance labeled PSI protocol that obtains BM25 term frequencies in a single execution. We comprehensively evaluate `Pisces` and find only a 1.87% deviation in retrieval accuracy relative to plaintext baselines. On large-scale datasets, our coarse-to-fine strategy reduces runtime by 41.21% and upload/download overhead by 68.77% compared to a fine-only strategy. Our proposed multi-instance labeled PSI further outperforms LSE by up to $496.03\times$ in runtime, $70733\times$ in upload overhead. These results demonstrate that `Pisces` is both accurate and efficient.

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

## A DETAILED PRELIMINARIES

### A.1 CRYPTOGRAPHIC PRIMITIVES

#### A.1.1 SECERT SHARING

Secret sharing (Shamir, 1979; Keller, 2020)is one of the critical primitives of MPC. In this paper, we adopt 2-out-of-2 arithmetic secret sharing technology. The main idea of it is to break a secret value into 2 shares, each of which is held by a party. For example, $\mathcal{S}$, who holds the secret value $x \in \mathbb{F}_p$, wants to secret share this secret value with another party $\mathcal{C}$. To do this, $P_S$ first generates a random value $r \in \mathbb{F}_p$ as its share $\langle x \rangle^S = r$, and then sends $\langle x \rangle^C = x - r \mod \mathbb{F}_p$ to another party $\mathcal{C}$. Therefore $x = \langle x \rangle^S + \langle x \rangle^C \mod \mathbb{F}_p$, which, for simplicity, we denote as $x = \langle x \rangle^S + \langle x \rangle^C$.

### A.1.2 LABELED PRIVATE SET INTERSECTION

The PSI (Jarecki & Liu, 2010) allows two parties, a server $\mathcal{S}$ and a client $\mathcal{C}$, to learn the intersection of their respective element sets without revealing any additional information outside the intersection. Labeled PSI (Chen et al., 2018; Bienstock et al., 2024; Cong et al., 2021) extends the traditional PSI by allowing the server $\mathcal{S}$ to associate a label with each element, and the client $\mathcal{C}$ learns the labels for elements in the intersection. Formally, $\mathcal{S}$ inputs a set of key-value pairs $\{(x_i, l(x_i))\}$, where $x_i$ is an element and $l(x_i)$ is its corresponding label, while $\mathcal{C}$ inputs a set of key $Y$. After execution, $\mathcal{C}$ learns a set of pairs $\{(y, l(y))\}$ for $y \in X \cap Y$.

### A.1.3 OBLIVIOUS PSEUDORANDOM FUNCTION

The OPRF (Freedman et al., 2005) is a cryptographic primitive that enables two parties, a server $\mathcal{S}$ and a client $\mathcal{C}$, to jointly compute a pseudorandom function (PRF) $F.(\cdot)$. As shown in Figure 3, $\mathcal{S}$ takes a PRF key $k$ as input and learns nothing, while $\mathcal{C}$ takes $x$ as input and learns the PRF value $F_k(x)$. Moreover, $\mathcal{C}$ learns nothing about the PRF key $k$ and $\mathcal{S}$ learns nothing about the input or the output of $\mathcal{C}$.

---

**Functionality $\mathcal{F}_{\mathrm{OPRF}}$**

**Parameters:** Two parties $\mathcal{S}$ and $\mathcal{C}$. A PRF $F.(\cdot)$.

**Functionality:**

- Wait for input $k$ from $\mathcal{S}$, where $k$ is a PRF key.
- Wait for input $x$ from $\mathcal{C}$.
- Output $F_k(x)$ to $\mathcal{C}$.

---

Figure 3: Ideal functionality of OPRF

### A.1.4 OBVIOUSLY KEY-VALUE STORE

The OKVS (Garimella et al., 2021) is a data structure that encodes a set of key-value pairs into a compact representation while preserving the privacy of both keys and values. The definition is as follows:

**Definition 1** (Oblivious Key-Value Store). *An OKVS parameterized by a key space $\mathcal{K}$ and a value $\mathcal{V}$ space, and consists of two algorithms:*

- $\Gamma$ *or* $\bot \leftarrow \mathrm{Encode}((k_1, v_1), (k_2, v_2), \ldots, (k_n, v_n))$*: The encode algorithm takes n key–value pairs* $\{(k_1, v_1), (k_2, v_2), \ldots, (k_n, v_n)\} \subset \{\mathcal{K} \times \mathcal{V}\}^n$ *as input, and outputs a structure* $\Gamma$ *(or an error terminator* $\bot$ *with negligible probability).*

- $v \leftarrow \mathrm{Decode}(\Gamma, k)$*: The decode algorithm takes an OKVS structure* $\Gamma$ *and a key* $k \in \mathcal{K}$ *as input, and outputs the corresponding value* $v \in \mathcal{V}$*.*

**Correctness:** An OKVS is correct if, for all $X \subset \mathcal{K} \times \mathcal{V}$ with distinct keys such that $\mathrm{Encode}(X) = \Gamma \neq \bot$ and $(k, v) \in X$, it holds that $\mathrm{Decode}(\Gamma, k) = v$;

**Computationally Obliviousness:** An OKVS is computationally oblivious if, for any two key sets with $n$ distinct keys $K = \{k_1, k_2, \ldots, k_n\} \subset \mathcal{K}$ and $K' = \{k'_1, k'_2, \ldots, k'_n\} \subset \mathcal{K}$ and a uniformly random value set $V = \{v_1, v_2, \ldots, v_n\} \subset \mathcal{V}$, a probabilistic polynomial-time adversary is not able to distinguish between $\mathrm{Encode}((k_1, v_1), (k_2, v_2), \ldots, (k_n, v_n)) = \Gamma \neq \bot$ and $\mathrm{Encode}((k'_1, v_1), (k'_2, v_2), \ldots, (k'_n, v_n)) = \Gamma' \neq \bot$.

This computationally obliviousness property ensures that the OKVS reveals no information about the encoded keys or values beyond the decoded results for given keys.

### A.1.5 BATCH PRIVATE INFORMATION RETRIEVAL-TO-SHARE

The PIR-to-share (Song et al., 2025) is a cryptographic primitive that enables a client $\mathcal{C}$ to privately retrieve the values corresponding to its queries from the server $\mathcal{S}$. After that, $\mathcal{S}$ and $\mathcal{C}$

obtain the secret shares of queried values, respectively. As shown in Figure 4, $\mathcal{S}$ takes its data $D$ of size $N$ as input, while $\mathcal{C}$ takes its queries $I = \{I_1, I_2, \ldots, I_b\}$ (index set) as input. $\mathcal{S}$ learns data shares $\langle D[I_1]\rangle^S, \langle D[I_2]\rangle^S, \ldots, \langle D[I_b]\rangle^S$ corresponding to $\mathcal{C}$'s queries and $\mathcal{C}$ learns data shares $\langle D[I_1]\rangle^C, \langle D[I_2]\rangle^C, \ldots, \langle D[I_b]\rangle^C$. During this process, $\mathcal{S}$ learns nothing about $\mathcal{C}$'s queries, and $\mathcal{C}$ only learns the secret shares of the retrieved values rather than the raw data of $\mathcal{S}$.

---

**Functionality $\mathcal{F}_{\text{PIR2Share}}$**

**Parameters:** Two parties $\mathcal{S}$ and $\mathcal{C}$.

**Functionality:**

- Wait for input $D$ from $\mathcal{S}$.
- Wait for input $I = \{I_1, I_2, \ldots, I_b\}$ from $\mathcal{C}$.
- Sample $\langle D[I_1]\rangle^S, \langle D[I_2]\rangle^S, \ldots, \langle D[I_b]\rangle^S$ and $\langle D[I_1]\rangle^C, \langle D[I_2]\rangle^C, \ldots, \langle D[I_b]\rangle^C$ uniformly, such that $\langle D[I_1]\rangle^S + \langle D[I_1]\rangle^C = \langle D[I_1]\rangle, \ldots, \langle D[I_b]\rangle^S + \langle D[I_b]\rangle^C = \langle D[I_b]\rangle$.
- Output the shares $\langle D[I_1]\rangle^S, \langle D[I_2]\rangle^S, \ldots, \langle D[I_b]\rangle^S$ to $P_S$ and $\langle D[I_1]\rangle^C, \langle D[I_2]\rangle^C, \ldots, \langle D[I_b]\rangle^C$ to $P_C$.

---

Figure 4: Ideal functionality of $\mathcal{F}_{\text{PIR2Share}}$

## A.2 SEMANTIC SIMILARITY

Semantic similarity (Awasthy et al., 2025; Zhang et al., 2025) is a measure of the degree to which the meanings of two linguistic units, such as words, phrases, sentences, or documents, are alike, based on their semantic content rather than lexical matching. It plays a fundamental role in many natural language processing tasks, including information retrieval and text summarization. Contemporary methods operationalize meaning via vector representations. Similarity is then measured with distance functions in embedding space, such as cosine similarity, Hamming distance, and Euclidean distance. In this paper, we choose cosine similarity as our similarity metric.

## A.3 BEST MATCHING 25

A popular algorithm to achieve lexical retrieval is BM25 (Robertson et al., 2009; Lù, 2024), which is a probabilistic information retrieval algorithm widely used to rank documents according to their relevance to a given query. It is an enhancement to the traditional term frequency-inverse document frequency (TF-IDF) algorithm, which measures the importance of a term within a set of documents. BM25 takes document length into account and introduces a saturation function to term frequencies, which helps prevent common terms from dominating the results to improve the ranking accuracy.

Given a document set $D = \{d_1, d_2, \ldots, d_N\}$ and a query $Q = \{q_1, q_2, \ldots, q_n\}$, where $d_i$ denotes the $i$-th document in $D$, $N$ is the total number of documents in $D$, $q_j$ is the $j$-th term in $Q$, $n$ is the total number of terms in $Q$, the BM25 relevance score for document $d_i$ relative to this query is defined as:

$$
\begin{aligned}
Score(Q, d_i) &= \sum_{j=1}^{n} \text{IDF}(q_j) \cdot R(q_j, d_i) \\
&= \sum_{j=1}^{n} \log\left(1 + \frac{N - df_j + 0.5}{df_j + 0.5}\right) \cdot \frac{tf_{i,j}}{tf_{i,j} + k_1 \cdot \left(1 - b + b \cdot \frac{L_{d_i}}{L_{ave}}\right)}
\end{aligned}
\tag{1}
$$

where $\text{IDF}(q_j)$ is the inverse document frequency of $q_j$ and $R(q_j, d)$ is the relevance score for the document $d_i$ relative to the term $q_j$. Besides, $df_j$ is the document frequency for term $q_j$, i.e. the number of documents in the document set $D$ in which $q_j$ appears, $tf_{i,j}$ is the term frequency of $q_j$ in the document $d_i$, $L_{d_i}$ is the length of the document $d_i$, $L_{ave}$ is the average length of the document set $D$, $k_1 > 0$ and $0 < b < 1$ are constant values, $k_1$ controls the saturation of the term frequency and $b$ adjusts the impact of normalization of document length.

## B NOTATION

We summarize the frequently used notation in Table 6.

Table 6: Notation Table

| Symbol | Description |
|---|---|
| $\mathcal{S}$ | The server, who holds a sensitive knowledge base. |
| $\mathcal{C}$ | The user, who holds a private query. |
| $D$ | Document set in the knowledge base. |
| $Q$ | Query. |
| $N$ | Chunk number of $D$. |
| $n$ | Unique token number of $Q$. |
| $c_i$ | $i$-th chunk. |
| $I_i$ | Index of chunk $c_i$. |
| $\mathbf{v}_i$ | Vector representation of chunk $c_i$. |
| $m$ | Unique token number of chunk $c_i$. |
| $w_{i,l}$ | $l$-th token of chunk $c_i$. |
| $tf_{i,l}$ | Term frequency of $w_{i,l}$ in chunk $c_i$. |
| $D^v = \{I_i;\ \mathbf{v}_i;\ c_i\}_{i \in [1,N]}$ | Embedded chunk set. |
| $D^t = \left\{I_i; \{w_{i,l} : tf_{i,l}\}_{l \in [1,m_i]}; c_i\right\}_{i \in [1,N]}$ | Tokenized chunk set. |
| $q_j$ | $j$-th token of query $Q$. |
| $\mathbf{q}$ | Vector representation of query $Q$. |
| $Q^t = \{q_1, q_2, \ldots, q_n\}$ | Tokenized query. |

## C  OBLIVIOUS FILTER

The core idea of the oblivious filter is to convert an approximate (fuzzy) matching problem into an exact matching task. The detailed oblivious filter protocol is shown in Protocol 3. In this protocol, both the server $\mathcal{S}$ and the user $\mathcal{C}$ select the same projections to mask their binary vector(s). A chunk is considered a candidate match if its projected binary vectors match the query's projected binary vectors on at least two projections. This protocol allows the server $\mathcal{S}$ to identify a candidate set of chunks that are likely to match the query.

To privately achieve the above matching, we employ a 2-out-of-T Shamir secret sharing scheme. The client $\mathcal{C}$ can only reconstruct a secret value if it obtains at least two shares for a chunk. Furthermore, to prevent the client to learn which specific chunks are matched, the server $\mathcal{S}$ encrypts all shares with additive HE. As a result, the client would reconstruct the secret in ciphertext, which ensures the client could not learn any information throughout the oblivious filter.

## D  MULTI-INSTANCE LABELED PRIVATE SET INTERSECTION

We design a customized multi-instance labeled PSI protocol (Protocol 4) to support repeated invocations of labeled PSI with the same small client query set. This protocol features two key innovations. First, the setup phase only involves the server $\mathcal{S}$ and produces a reusable OKVS structure $\Gamma$. It can be efficiently reused across multiple queries without recomputation. Second, the interactive phase minimizes computational overhead. It only requires a single, small-scale OPRF execution per query, independent of the server's data size. These optimizations significantly reduce both communication and computation costs compared to conventional labeled PSI protocols.

## E  DETAILED DATASET

Table 7 shows the details of the datasets used in this paper, including the number of documents and the number of chunks.

---

**Protocol 3:** $\prod_{\text{Oblivious\_Filter}}$

**Input:** $\mathcal{S}$ inputs the set $D^v = \left\{(I_i, \mathbf{v}_i^{\mathbf{b}}, c_i)\right\}_{i \in [1,N]}$, where $I_i$ is the binary vector representation corresponding to the chunk $c_i$. $\mathcal{C}$ inputs binary vector $\mathbf{q}^{\mathbf{b}} \in \{0,1\}^{\mathcal{L}}$.

**Output:** $\mathcal{S}$ learns a candidate set $D' = \left\{(I_i', \mathbf{v}_i^{\mathbf{b}'}, c_i')\right\}_{i \in [1,N']}$, where for all $i \in [1, N']$, $\mathsf{HD}(\mathbf{v}_i^{\mathbf{b}'}, \mathbf{q}^{\mathbf{b}}) \leq t$ (i.e., Hamming distance at most $t$).

**Setup Phase:**

1: $\mathcal{S}$ generates a random keypair $(pk, sk)$ for an additive HE scheme.
2: $\mathcal{S}$ sets the projection weight $\ell \leftarrow \lceil \sqrt{t \cdot \mathcal{L}} \rceil$ and the number of projections $T \leftarrow 160$. $\mathcal{S}$ randomly selects $T$ projection masks $\{\mathbf{m}_i \in \{0,1\}^{\mathcal{L}}\}_{i \in [1,T]}$ such that $\|\mathbf{m}_i\| = \ell$ for all $i \in [1, T]$.
3: $\mathcal{S}$ selects $2N$ random numbers $\{x_i\}_{i \in [1,N]}$ and $\{s_i\}_{i \in [1,N]}$, and initializes an empty collection $\mathbb{C}$.
4: $\mathcal{S}$ selects a random linear polynomial $P_i(x) = ax + s_i$ with random coefficient $a$ for $i \in [1, N]$.
5: $\mathcal{S}$ computes ciphertext $v_{i,j} \leftarrow \mathsf{Enc}(pk, P_i(x_j))$ and key $k_{i,j} \leftarrow \mathsf{Hash}(\mathbf{v}_i^{\mathbf{b}} \wedge \mathbf{m}_j)$ for $i \in [1,N], j \in [1,T]$.
6: $\mathcal{S}$ inserts the pair $(k_{i,j}, v_{i,j})$ into $\mathbb{C}$.
7: $\mathcal{S}$ invokes $\mathsf{OKVS.Encode}(\mathbb{C})$ to obtain the OKVS structure $\Gamma$.

**Interactive Phase:**

1: $\mathcal{S}$ sends the public key $pk$, projection masks $\{\mathbf{m}_i\}_{i \in [1,T]}$, OKVS structure $\Gamma$, and random numbers $\{x_i\}_{i \in [1,T]}$ to $\mathcal{C}$.
2: $\mathcal{C}$ computes $t_j \leftarrow \mathsf{Hash}(\mathbf{q}^{\mathbf{b}} \wedge \mathbf{m}_j)$ for $j \in [1,T]1$.
3: $\mathcal{C}$ invokes $\mathsf{OKVS.Decode}(\Gamma, \{t_j\}_{j \in [1,T]})$ to obtain $\{d_i\}_{i \in [1,T]}$.
4: $\mathcal{C}$ computes a candidate secret ciphertext: $s_{i,j} \leftarrow d_j - x_j \cdot \frac{d_i - d_j}{x_i - x_j}$ for each combination $(i, j)$ from the $\binom{T}{2}$ possible pairs of indices from $[1, T]$.
5: $\mathcal{C}$ shuffles all computed ciphertexts $\{s_{i,j}\}$ to form the set $\mathbb{S}$ and sends $\mathbb{S}$ to $\mathcal{S}$.
6: $\mathcal{S}$ decrypts $\mathbb{S}$ to obtain $\mathbb{P} \leftarrow \{\mathsf{Dec}(sk, s) \mid s \in \mathbb{S}\}$.
7: For each $s_i$ (from the original setup) that appears in $\mathbb{P}$, $\mathcal{S}$ adds the corresponding item $(I_i, \mathbf{v}_i^{\mathbf{b}}, c_i)$ to the result set $D'$.
8: $\mathcal{S}$ returns $D'$ as the final result.

---

Table 7: Details of datasets we evaluated in this paper. "Documents" denotes the number of documents in the dataset, and "Chunks" denotes the number of chunks generated from breaking down all the documents in the dataset.

| | Dataset | Documents | Chunks |
|---|---|---|---|
| | Dev_answerable | 290 | 1990 |
| ClapNQ | Train_answerable | 1751 | 14010 |
| | Train_single_answerable | 8996 | 71363 |
| SQuAD | Dev_v2.0 | 35 | 1204 |
| | Training_v2.0 | 442 | 19029 |
| | Dev_distractor | 66581 | 269602 |
| HotpotQA | Dev_fullwiki | 66573 | 276013 |
| | Training | 482021 | 1795146 |

## F  SUPPLEMENTARY ACCURACY EXPERIMENTAL RESULTS

The top-10 retrieval accuracy comparison between `Pisces` and plaintext baseline is shown in Figure 5. Furthermore, we evaluate the retrieval accuracy of our proposed `Pisces` against existing DP-based approaches listed in Table 1. The detailed top-5 and top-10 accuracy results, presented in

---

**Protocol 4:** $\prod_{\mathsf{MultLPSI}}$

**Input:** $\mathcal{S}$ inputs set $D^t = \{w_{i,l} : tf_{i,l}\}_{i \in [1,N], l \in [1,m_i]}$. $\mathcal{C}$ inputs set $Q^t = \{q_1, q_2, \ldots, q_n\}$.

**Output:** $\mathcal{C}$ learns $\{tf'_{i,j}\}_{i \in [1,N]], j \in [1,n]}$, where if $q_j = w_{i,l}$ then $tf'_{i,j} = tf_{i,l}$, and otherwise $tf'_{i,j} = 0$.

**Setup Phase:**

1: $\mathcal{S}$ selects a random PRF key $k$ and two key derivation functions $\mathsf{KDF}_0$ and $\mathsf{KDF}_1$.
2: $\mathcal{S}$ initializes an empty set $\mathbb{S}$.
3: $\mathcal{S}$ computes $r_{i,l} \leftarrow \mathsf{PRF}(k, w_{i,l})$, $k_{i,l} \leftarrow \mathsf{KDF}_0(i, r_{i,l})$, $m_{i,l} \leftarrow \mathsf{KDF}_1(i, r_{i,l})$ and $c_{i,l} \leftarrow \mathsf{AES.Enc}(m_{i,l}, 0^\ell \| tf_{i,l})$ for $i \in [1,N], l \in [1,m_i]$.
4: $\mathcal{S}$ inserts the key-value pair $(k_{i,l}, c_{i,l})$ into $\mathbb{S}$ for $i \in [1,N], l \in [1,m_i]$.
5: $\mathcal{S}$ invokes $\mathsf{OKVS.Encode}(\mathbb{S})$ to obtain the OKVS structure $\Gamma$.

**Interactive Phase:**

1: $\mathcal{S}$ sends the OKVS structure $\Gamma$ and key derivation functions $\mathsf{KDF}_0$, $\mathsf{KDF}_1$ to $\mathcal{C}$.
2: $\mathcal{S}$ and $\mathcal{C}$ invoke an OPRF protocol with PRF key $k$ and $Q^t = \{q_1, q_2, \ldots, q_n\}$ as inputs, respectively. After execution, $\mathcal{C}$ obtains the PRF results $\mathbb{D} = \{d_1, d_2, \ldots, d_n\}$.
3: $\mathcal{C}$ initializes $\mathbb{K}_i = \emptyset$ and $\mathbb{M}_i = \emptyset$ for $i \in [1,N]$.
4: $\mathcal{C}$ computes $k'_{i,j} \leftarrow \mathsf{KDF}_0(i, d_j)$ and $m'_{i,j} \leftarrow \mathsf{KDF}_1(i, d_j)$ for $i \in [1,N], j \in [1,n]$.
5: $\mathcal{C}$ adds $k'_{i,j}$ to $\mathbb{K}_i$ and $m'_{i,j}$ to $\mathbb{M}_i$ for $i \in [1,N], j \in [1,n]$.
6: $\mathcal{C}$ invokes $\mathsf{OKVS.Decode}(\Gamma, \mathbb{K}_i)$ to obtain ciphers $\{c_{i,j}\}_{i \in [1,N], j \in [1,n]}$.
7: $\mathcal{C}$ computes $p_{i,j} \leftarrow \mathsf{AES.Dec}(m'_{i,j}, c_{i,j})$ for $i \in [1,N], j \in [1,n]$.
8: If $p_{i,j}$ starts with $0^\ell$ (where $\ell$ is a security parameter), $\mathcal{C}$ parses $p_{i,j}$ as $0^\ell \| v_{i,j}$ and set $tf'_{i,j} \leftarrow v_{i,j}$. Otherwise, set $tf'_{i,j} \leftarrow 0$.
9: $\mathcal{C}$ returns $\{tf'_{i,j}\}_{i \in [1,N], j \in [1,n]}$ as the result.

---

Table 8 and Table 9 respectively. Following the original papers, we configure the privacy parameters as: $\varepsilon = 1$ for DP-RAG (Grislain, 2025), $\varepsilon = 1280$ for RemoteRAG (Cheng et al., 2024), and $\sigma = 0.1$ for (Yao & Li, 2025).

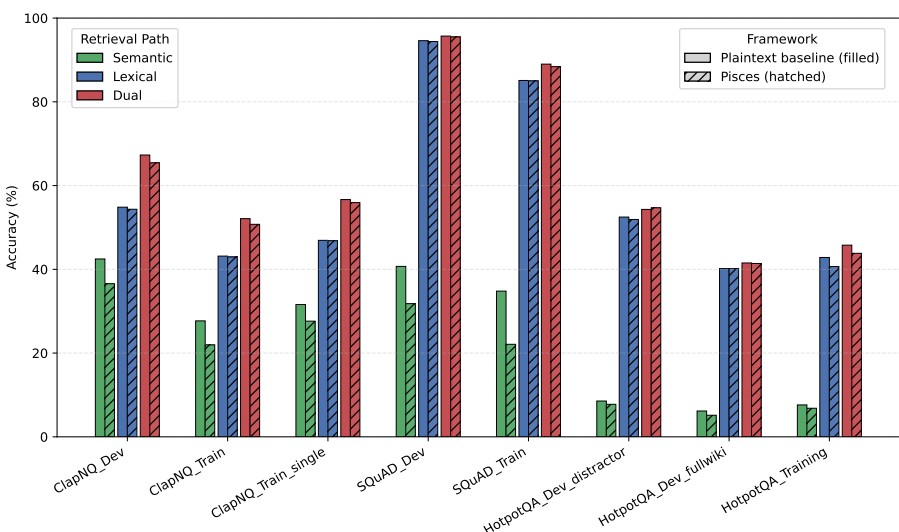

Figure 5: Top-10 retrieval accuracy comparison between `Pisces` and plaintext baseline over ground-truth.

Table 8: Top-5 retrieval accuracy comparison between `Pisces` and baselines over ground-truth. Results for DP-RAG (Grislain, 2025) uses $\varepsilon = 1$, RemoteRAG (Cheng et al., 2024) uses $\varepsilon = 1280$, and (Yao & Li, 2025) uses $\sigma = 0.1$.

| Dataset | | Framework | Top-5 | | |
|---|---|---|---|---|---|
| | | | Semantic | Lexical | Dual-Path |
| ClapNQ | Dev_answerable | Plaintext | 36.38% | 47.42% | 59.66% |
| | | DP-RAG (Grislain, 2025) | 26.37% | - | 26.37% |
| | | RemoteRAG (Cheng et al., 2024) | 35.54% | - | 35.54% |
| | | (Yao & Li, 2025) | 15.45% | - | 15.45% |
| | | Pisces | 30.14% | 47.15% | **58.05%** |
| | Train_answerable | Plaintext | 22.99% | 36.78% | 45.78% |
| | | DP-RAG (Grislain, 2025) | 17.47% | - | 17.47% |
| | | RemoteRAG (Cheng et al., 2024) | 22.39% | - | 22.39% |
| | | (Yao & Li, 2025) | 6.68% | - | 6.68% |
| | | Pisces | 19.22% | 36.72% | **44.41%** |
| | Train_single_answerable | Plaintext | 27.57% | 40.12% | 50.56% |
| | | DP-RAG (Grislain, 2025) | 21.31% | - | 21.31% |
| | | RemoteRAG (Cheng et al., 2024) | 26.52% | - | 26.52% |
| | | (Yao & Li, 2025) | 4.41% | - | 4.41% |
| | | Pisces | 23.44% | 39.82% | **49.39%** |
| SQuAD | Dev_v2.0 | Plaintext | 33.80% | 91.90% | 93.30% |
| | | DP-RAG (Grislain, 2025) | 24.74% | - | 24.74% |
| | | RemoteRAG (Cheng et al., 2024) | 33.18% | - | 33.18% |
| | | (Yao & Li, 2025) | 14.78% | - | 14.78% |
| | | Pisces | 25.20% | 91.60% | **93.10%** |
| | Training_v2.0 | Plaintext | 26.4% | 81.10% | 84.80% |
| | | DP-RAG (Grislain, 2025) | 20.34% | - | 20.34% |
| | | RemoteRAG (Cheng et al., 2024) | 25.48% | - | 25.48% |
| | | (Yao & Li, 2025) | 9.08% | - | 9.08% |
| | | Pisces | 17.40% | 80.90% | **84.10%** |
| HotpotQA | Dev_distractor | Plaintext | 7.28% | 43.62% | 46.27% |
| | | DP-RAG (Grislain, 2025) | 5.70% | - | 5.70% |
| | | RemoteRAG (Cheng et al., 2024) | 7.09% | - | 7.09% |
| | | (Yao & Li, 2025) | 1.27% | - | 1.27% |
| | | Pisces | 6.57% | 43.60% | **46.08%** |
| | Dev_fullwiki | Plaintext | 4.99% | 34.86% | 36.44% |
| | | DP-RAG (Grislain, 2025) | 3.87% | - | 3.87% |
| | | RemoteRAG (Cheng et al., 2024) | 4.97% | - | 4.97% |
| | | (Yao & Li, 2025) | 1.02% | - | 1.02% |
| | | Pisces | 4.36% | 34.80% | **36.00%** |
| | Training | Plaintext | 6.69% | 36.24% | 39.21% |
| | | DP-RAG (Grislain, 2025) | 4.00% | - | 4.00% |
| | | RemoteRAG (Cheng et al., 2024) | 6.48% | - | 6.48% |
| | | (Yao & Li, 2025) | 0.83% | - | 0.83% |
| | | Pisces | 6.01% | 34.42% | **37.74%** |

Table 9: Top-10 retrieval accuracy comparison between `Pisces` and baselines over ground-truth. Results for DP-RAG (Grislain, 2025) uses $\varepsilon = 1$, RemoteRAG (Cheng et al., 2024) uses $\varepsilon = 1280$, and (Yao & Li, 2025) uses $\sigma = 0.1$.

| Dataset | | Framework | Top-10 | | |
|---|---|---|---|---|---|
| | | | Semantic | Lexical | Dual-Path |
| ClapNQ | Dev_answerable | Plaintext | 42.47% | 54.84% | 67.30% |
| | | DP-RAG (Grislain, 2025) | 36.92% | - | 36.92% |
| | | RemoteRAG (Cheng et al., 2024) | 41.91% | - | 41.91% |
| | | (Yao & Li, 2025) | 20.63% | - | 20.63% |
| | | `Pisces` | 36.56% | 54.35% | **65.43%** |
| | Train_answerable | Plaintext | 27.69% | 43.16% | 52.10% |
| | | DP-RAG (Grislain, 2025) | 23.79% | - | 23.79% |
| | | RemoteRAG (Cheng et al., 2024) | 26.92% | - | 26.92% |
| | | (Yao & Li, 2025) | 9.04% | - | 9.04% |
| | | `Pisces` | 21.95% | 42.96% | **50.75%** |
| | Train_single_answerable | Plaintext | 31.60% | 46.92% | 56.66% |
| | | DP-RAG (Grislain, 2025) | 27.94% | - | 27.94% |
| | | RemoteRAG (Cheng et al., 2024) | 30.33% | - | 30.33% |
| | | (Yao & Li, 2025) | 6.05% | - | 6.05% |
| | | `Pisces` | 27.63% | 46.83% | **55.92%** |
| SQuAD | Dev_v2.0 | Plaintext | 40.70% | 94.60% | 95.70% |
| | | DP-RAG (Grislain, 2025) | 34.86% | - | 34.86% |
| | | RemoteRAG (Cheng et al., 2024) | 40.00% | - | 40.00% |
| | | (Yao & Li, 2025) | 19.94% | - | 19.94% |
| | | `Pisces` | 31.80% | 94.40% | **95.50%** |
| | Training_v2.0 | Plaintext | 34.80% | 85.10% | 89.00% |
| | | DP-RAG (Grislain, 2025) | 30.12% | - | 30.12% |
| | | RemoteRAG (Cheng et al., 2024) | 30.12% | - | 30.12% |
| | | (Yao & Li, 2025) | 12.20% | - | 12.20% |
| | | `Pisces` | 22.10% | 85.00% | **88.40%** |
| HotpotQA | Dev_distractor | Plaintext | 8.55% | 52.48% | 54.69% |
| | | DP-RAG (Grislain, 2025) | 7.45% | - | 7.45% |
| | | RemoteRAG (Cheng et al., 2024) | 8.52% | - | 8.52% |
| | | (Yao & Li, 2025) | 1.84% | - | 1.84% |
| | | `Pisces` | 7.77% | 51.91% | **54.31%** |
| | Dev_fullwiki | Plaintext | 6.17% | 40.18% | 41.52% |
| | | DP-RAG (Grislain, 2025) | 5.45% | - | 5.45% |
| | | RemoteRAG (Cheng et al., 2024) | 6.08% | - | 6.08% |
| | | (Yao & Li, 2025) | 1.30% | - | 1.30% |
| | | `Pisces` | 5.16% | 40.17% | **41.40%** |
| | Training | Plaintext | 7.62% | 42.83% | 45.77% |
| | | DP-RAG (Grislain, 2025) | 5.50% | - | 5.50% |
| | | RemoteRAG (Cheng et al., 2024) | 7.40% | - | 7.40% |
| | | (Yao & Li, 2025) | 1.22% | - | 1.22% |
| | | `Pisces` | 6.84% | 40.66% | **43.83%** |

