# OpenReview forum: "Pisces: Cryptography-based Private Retrieval-Augmented Generation with Dual-Path Retrieval"
_ICLR.cc/2026/Conference — ICLR 2026 Poster_

### Official Review · Reviewer_Nqrz · 2025-10-29

**Soundness:** 1
**Presentation:** 2
**Contribution:** 3
**Rating:** 4
**Confidence:** 5

**Summary:**

The paper focuses on an important research question of privacy-preserving retrieval augmented generation. Compared to differential privacy based methods, cryptography methods can provide more utility and stronger privacy guarantees with a scarification of efficiency and a burden for users to upload and download more data. Authors show that the feasibility of the solution with experiments

**Strengths:**

1. The paper fills the blanks of no cryptography solution in the field of privacy-preserving RAGs.
2. The paper considers the case of semantic and lexical retrieval.

**Weaknesses:**

1. Authors mentioned some DP-based methods and argued that they may harm utility. Is there any empirical experiment to support that? It seems that I didn't find a comparison to these baselines in the evaluation.
2. The motivation is confused. Why the answer to "How can we maintain retrieval performance while ensuring privacy for both the query and documents during retrieval?" Is using cryptography-based methods? Why do other methods not achieve this?

I think the motivation and high-level idea is interesting but there are some technical problems that need to be resolved. I hope the authors can provide some rationale for them before the paper reaches the acceptance standard.

**Questions:**

1. I think work in Huang et al., 2023 and Yao & Li, 2025 considered lexical retrieval. But they did not consider the retrieval of considering semantic and lexical retrieval at the same time. Is this a point that the paper wants to articulate?
2. I think this baseline is missing. [1]
3. I am a little bit confused by the structure. So, at the final inference stage, retrieved documents and queries are encrypted through MCP and the user uses a private key to decode and get responses. But during the retrieval phase, the tokenizer and embedding model need the plain text of the query, would that leak privacy? I think at this stage, the query is not encrypted.
4. Why in Figure 2, ClapNQ_Train_single. Pisces (with MPC) has higher accuracy than plaintext? Is it because of randomness? If so, I think multiple experiments should be conducted and error bars need to be reported.
5. Why in Table 4, HopotQA, encrypted text computation can have less time than plain text computation?

[1] Koga, Tatsuki, Ruihan Wu, and Kamalika Chaudhuri. "Privacy-Preserving Retrieval-Augmented Generation with Differential Privacy." arXiv preprint arXiv:2412.04697 (2024).

---

> ### Author Response · Authors · 2025-11-25
> **Response to Reviewer Nqrz [1/2]**
>
> Dear Reviewer Nqrz,
>
> We sincerely thank the reviewer for your time and insightful feedback. We are grateful for the positive assessment of our work and the constructive comments, which help us significantly improve our paper. **All comments have been carefully addressed, and the corresponding revisions have been updated in the revised paper with major changes highlighted in purple for clarity**. We hope our revisions address all the concerns and bring the paper up to acceptance.
>
> Below, we provide point-by-point responses to each comment, along with additional explanations and experimental results where necessary.
>
> *  *Weakness 1. Lack of comparison against DP-based methods*
>
> **Response to Weakness 1.**
> Thank you for your questions.
> We have added experimental comparisons (Table 8 in Appendix B.7) against DP-based methods in our revised paper to support our claim that DP-based methods may harm utility.
> Experimental results show that our accuracy is significantly higher than that of DP-based works. As for RemoteRAG listed in Table 1, we will complete the experimental results and update them as soon as possible (before 03/12/2025).  Below, we show part of the experimental results, and please refer to the revised paper for the complete results.
>
> Table 8 Retrieval accuracy comparisons between Pisces and baselines over ground-truth.
>
> | Dataset            |          | Framework           | Top-5 Semantic | Top-5 Lexical | Top-5 Dual-path | Top-10 Semantic | Top-10 Lexical | Top-10 Dual-path |
> |--------------------|---------------|---------------------|----------------|---------------|-----------------|-----------------|----------------|------------------|
> | **ClapNQ**         | Dev           | Plaintext           | 36.38%         | 47.42%        | 59.66%          | 42.47%          | 54.84%         | 67.30%           |
> |                    | Dev           | Pisces              | 30.14%         | 47.15%        | 58.05%          | 36.56%          | 54.35%         | 65.43%           |
> |                    | Dev           | DP-RAG              | 26.37%         | /             | 26.37%          | 36.92%          | /              | 36.92%           |
> |                    | Dev           | RandomProjection    | 15.45%         | /             | 15.45%          | 20.63%          | /              | 20.63%           |
> |                    | Train         | Plaintext           | 22.99%         | 36.78%        | 45.78%          | 27.69%          | 43.16%         | 52.10%           |
> |                    | Train         | Pisces              | 19.22%         | 36.72%        | 44.41%          | 21.95%          | 42.96%         | 50.75%           |
> |                    | Train         | DP-RAG              | 17.47%         | /             | 17.47%          | 23.79%          | /              | 23.79%           |
> |                    | Train         | RandomProjection    | 6.68%          | /             | 6.68%           | 9.04%           | /              | 9.04%            |
> |                    | Train_single  | Plaintext           | 27.57%         | 40.12%        | 50.56%          | 31.60%          | 46.92%         | 56.66%           |
> |                    | Train_single  | Pisces              | 23.44%         | 39.82%        | 49.39%          | 27.63%          | 46.83%         | 55.92%           |
> |                    | Train_single  | DP-RAG              | 21.31%         | /             | 21.31%          | 27.94%          | /              | 27.94%           |
> |                    | Train_single  | RandomProjection    | 4.41%          | /             | 4.41%           | 6.05%           | /              | 6.05%            |
> | **SQuAD**          | Dev           | Plaintext           | 33.80%         | 91.90%        | 93.30%          | 40.70%          | 94.60%         | 95.70%           |
> |                    | Dev           | Pisces              | 25.20%         | 91.60%        | 93.10%          | 31.80%          | 94.40%         | 95.50%           |
> |                    | Dev           | DP-RAG              | 24.74%         | /             | 24.74%          | 34.86%          | /              | 34.86%           |
> |                    | Dev           | RandomProjection    | 14.78%         | /             | 14.78%          | 19.94%          | /              | 19.94%           |
> |                    | Train         | Plaintext           | 26.40%         | 81.10%        | 84.80%          | 34.80%          | 85.10%         | 89.00%           |
> |                    | Train         | Pisces              | 17.40%         | 80.90%        | 84.10%          | 22.10%          | 85.00%         | 88.40%           |
> |                    | Train         | DP-RAG              | 20.34%         | /             | 20.34%          | 30.12%          | /              | 30.12%           |
> |                    | Train         | RandomProjection    | 9.08%          | /             | 9.08%           | 12.20%          | /              | 12.20%           |

---

> ### Author Response · Authors · 2025-11-25
> **Response to Reviewer Nqrz [2/2]**
>
> *  *Weakness 2. Confusing motivation of using cryptography-based methods vs other methods*
>
> **Response to Weakness 2.**
> Cryptography is the only solution that can simultaneously (1) hide queries and documents during the retrieval process, (2) limit trust in any single party, and (3) support dual-path retrieval.
> Cryptography-based technologies (such as MPC) could support not only the similarity computations required by semantic retrieval, but also the exact term matching essential for lexical retrieval, addressing the key limitations of DP-based approaches.
> This is because cryptography-based technologies do not alter the raw data, while DP involves perturbing the data with irreversible noise, which makes it difficult to perform exact matching.
>
>
> *  *Question 1. Articulating supporting both semantic and lexical retrieval*
>
> **Response to Question 1.**
> Indeed, this is the point of the paper where we show that we can support both semantic and lexical retrieval.
> These works actually considered semantic retrieval only, as we replied in the **Response to Weakness 2**, DP-based approaches are difficult to support lexical retrieval.
>
> *  *Question 2. Missing baseline*
>
> **Response to Question 2.**
> The goal of [1] is different from ours. We aim to protect the privacy of the retrieval process in the RAG system, while [1] aims to protect the privacy of the generation process in the RAG system.
> Besides, Pisces can be integrated with [1] to achieve end-to-end privacy in the RAG system, and the reference [1] is already included in our paper in Section 4.4 (reference (Koga et al., 2024)).
>
> [1] Koga, Tatsuki, Ruihan Wu, and Kamalika Chaudhuri. "Privacy-Preserving Retrieval-Augmented Generation with Differential Privacy." arXiv preprint arXiv:2412.04697 (2024).
>
> *  *Question 3. Confused structure*
>
> **Response to Question 3.**
> This is a valid and important question. It would not leak privacy as the tokenizer and embedding model are deployed locally with the user. Thus, we do not need to encrypt them at this point. We have updated Figure 1 to better illustrate this point in our revised paper.
>
> *  *Question 4. Higher accuracy for some experiment results*
>
> **Response to Question 4.**
> Thank you for identifying clarity issues in our paper.
> We used the open-source library ChromaDB to implement the plaintext RAG system. However, we found that ChromaDB uses the Approximate Nearest Neighbours search (ANNs) for ranking similarities, which resulted in non-deterministic results. In our revised paper, we update the experimental results by using an exact ranking. The revised results (Figure 2) show that the plaintext baseline has a higher accuracy than our proposed framework.
>
>
> *  *Question 5. Smaller computation time for the encrypted compared to the plaintext*
>
> **Response to Question 5.**
> Table 4 is actually a comparison between a fine-only strategy and a coarse-to-fine strategy, where both strategies are implemented with encrypted computations. For a large-scale dataset, our coarse-to-fine strategy requires a significantly smaller retrieval time compared to the fine-only strategy.

---

> > ### Comment · Reviewer_Nqrz · 2025-11-25
> >
> > Thank you for the response and extra response. I have read other reviews and your responses as well. I think most answers to my questions make sense.
> >
> > A follow-up question regarding Question 2:
> > 1. What is the major benefit of using privacy-preserving techniques at retrieval phase. I see the difference that you encrypt docs at retrieval while [1] do not protect any privacy at retrieval. [1] mainly uses DP-aggregation at final generation step. One benefit is that your method protects user queries as well. But let's stand on the point of preventing the server knowledge base from being retrieved by malicious queries, will a cryptography based method bring any extra advantage except more affirmative privacy-preserving reliability? (e.g. accuracy, utility, generation performance, retrieval efficiency?)

---

> > > ### Author Response · Authors · 2025-11-27
> > > **Response to Reviewer Nqrz**
> > >
> > > Dear Reviewer Nqrz,
> > >
> > > We sincerely thank you for your quick response and for adjusting the rating. We appreciate that our responses have addressed most of your queries.
> > >
> > > Below is the response to the follow-up question regarding Question 2.
> > >
> > > This is an excellent question. The major benefit of cryptography-based methods over DP-based methods lies in the ability to preserve retrieval accuracy and utility similar to plaintext levels. Additionally, cryptography-based methods provide privacy protection during execution, i.e., the retrieval computation, rather than the output[1]. This means that cryptography-based methods cannot prevent the server's knowledge base from being retrieved by malicious queries. We will consider a hybrid solution using cryptography and DP in future work.
> > >
> > > [1] Meisingseth F, Rechberger C. SoK: Computational and Distributed Differential Privacy for MPC[J]. Proceedings on Privacy Enhancing Technologies, 2025.
> > >
> > > Thank you again for taking the time to re-evaluate our paper after reading our responses.

---

> > > > ### Comment · Reviewer_Nqrz · 2025-11-27
> > > >
> > > > Makes sense.

---

### Official Review · Reviewer_EtQD · 2025-10-30

**Soundness:** 3
**Presentation:** 3
**Contribution:** 3
**Rating:** 6
**Confidence:** 3

**Summary:**

Pisces proposes the first cryptography-based retrieval-augmented generation (RAG) framework that simultaneously protects both user queries and documents through dual-path (semantic and lexical) private retrieval, achieving near-plaintext accuracy with significantly improved efficiency.

**Strengths:**

1. The paper presents a well-structured and logically coherent framework.

2. The proposed method designs customized cryptographic protocols that effectively balance privacy protection and retrieval efficiency.

**Weaknesses:**

1. Limited support for multi-user concurrency. The current framework is designed for a single client–server setting and does not easily extend to multi-user or concurrent query scenarios.

2. Limited dataset scale and diversity. The evaluation is conducted on moderate-sized, English-only datasets, which limits the generalizability to large-scale or domain-specific applications.

3. Lack of empirical security analysis. The paper provides theoretical privacy guarantees but does not include empirical evaluations against potential attacks such as reconstruction or traffic analysis.

**Questions:**

See weakness

---

> ### Author Response · Authors · 2025-11-25
> **Response to Reviewer EtQD**
>
> Dear Reviewer EtQD,
>
> We sincerely thank the reviewer for your time and insightful feedback. We are grateful for the positive assessment of our work and the constructive comments, which help us significantly improve our paper. **All comments have been carefully addressed, and the corresponding revisions have been updated in the revised paper with major changes highlighted in purple for clarity**. We hope our revisions address all the concerns and bring the paper up to acceptance.
>
>
> Below, we provide point-by-point responses to each comment, along with additional explanations and experimental results where necessary.
>
>
> *  *Weakness 1. Limited support for multi-user concurrency*
>
> **Response to Weakness 1.**
> Our framework can support multi-user concurrency. Each user can run a separate instance with the server for its private retrieval process with our proposed framework. The chunks used for the lexical retrieval and the embeddings for the sementic retrival can be used for all users as they would remain the same for any RAG queries.
>
>
> *  *Weakness 2. Limited dataset scale*
>
> **Response to Weakness 2.**
> We appreciate the suggestion, and we have added more experiments about a large-scale dataset (hotpotqa_training with 1,795,146 chunks) in Table 2, Table 3, Table 8, Figure 2, and Figure 5 of our revised paper.
> Experimental results demonstrate that even with a large-scale dataset, our proposed framework can achieve high retrieval accuracy.
> Below, we provide the experimental results in Table 2 as an example.
>
> Table 2: Semantic retrieval accuracy against the plaintext baseline.
>
> | Dataset      | Split            | Top-5 Accuracy | Top-5 Time | Top-10 Accuracy | Top-10 Time |
> |--------------|------------------|----------------|------------|-----------------|-------------|
> | **ClapNQ**   | Dev              | 87.67%         | 3.47       | 86.83%          | 3.56        |
> |              | Train            | 80.30%         | 4.12       | 77.78%          | 4.17        |
> |              | Train_single     | 90.44%         | 7.33       | 81.39%          | 7.88        |
> | **SQuAD**    | Dev              | 78.02%         | 3.37       | 75.96%          | 3.41        |
> |              | Train            | 78.14%         | 4.38       | 74.86%          | 4.46        |
> | **HotpotQA** | Dev_distractor   | 79.90%         | 18.91      | 79.80%          | 20.10       |
> |              | Dev_fullwiki     | 79.46%         | 19.27      | 78.23%          | 20.76       |
> |              | **Training**         | **82.92%**         | **147.08**     | **81.42%**         | **160.90**      |
>
> *  *Weakness 3. Lack of empirical security analysis*
>
> **Response to Weakness 3.**
> We consider a semi-honest threat model, where the parties involved follow the protocol specification honestly, while trying to obtain information about others. Given this setting, we consider these potential attacks to be out of scope.

---

### Official Review · Reviewer_24E7 · 2025-11-01

**Soundness:** 2
**Presentation:** 2
**Contribution:** 1
**Rating:** 4
**Confidence:** 5

**Summary:**

The paper proposes Pisces, the first practical cryptography-based RAG framework that supports dual-path retrieval, while protecting both the query and documents.

Along the semantic retrieval path, we reduce computation and communication overhead by leveraging a coarse-to-fine strategy. Specifically, a novel oblivious filter is used to privately select a candidate set of documents to reduce the scale of subsequent cosine similarity computations.

**Strengths:**

The paper proposes the first cryptography-based RAG retrieval framework with dual-path retrieval, while
ensuring privacy for both the query and documents.

The authors  propose a coarse-to-fine strategy for the semantic retrieval path with an oblivious filter to
reduce computation and communication complexity.

**Weaknesses:**

1. The motivation of why applying MPC to RAG with LLMs is seriously unclear to me.Please rewrite this part.


2. The motivation and possible usage scenarios of your proposed system in reality  needs to be verified.

3. Please provide the strict privacy definition, your Cryptography-based Private RAG seems only protect the privacy in the whole system in certain parts.

**Questions:**

see the above comments.

---

> ### Author Response · Authors · 2025-11-25
> **Response to Reviewer 24E7**
>
> Dear Reviewer 24E7,
>
> We sincerely thank the reviewer for your time and insightful feedback. We are grateful for the positive assessment of our work and the constructive comments, which help us significantly improve our paper. **All comments have been carefully addressed, and the corresponding revisions have been updated in the revised paper with major changes highlighted in purple for clarity**. We hope our revisions address all the concerns and bring the paper up to acceptance.
>
>
> Below, we provide point-by-point responses to each comment, along with additional explanations and experimental results where necessary.
>
>
> *  *Weakness 1. Motivation for applying MPC to RAG.*
>
> *  *Weakness 2. Verification of motivation and possible usage scenarios.*
>
>
> **Response to Weaknesses 1 and 2.**
> Thank you for identifying clarity issues in our paper.
> We address these concerns in two aspects: (1) we update our motivating example to highlight the privacy concerns in RAG systems, and (2) the difference between cryptography-based and DP-based approaches.
>
> (1) The RAG system often interacts with highly sensitive personal data, which leads to significant privacy concerns.
> For instance, in personalised healthcare diagnostics, a healthcare agent with an LLM as the cornerstone for which personalised queries regarding one's health are made against, supported by an RAG system that encompasses a medical knowledge base. The user interacting with the agent would be highly concerned about revealing too much personal information, such as family history, for fear of any potential privacy exposure.
> Simultaneously, queries could reveal information about the individuals part of the RAG knowledge base, which contains highly sensitive personal information that is hard to anonymize in nature, such as rare diseases, clinical notes, or biometric identifiers. These works [1, 2] demonstrate that it is possible to extract sentences verbatim or personally identifiable information from the knowledge base.
> The leakage of any sensitive information would violate data privacy regulations, such as GDPR, PIPL, and HIPAA. This highlights the need for privacy in a reciprocal manner, where, during retrieval, the knowledge base does not learn additional information about the user, and the user does not learn additional information about the knowledge base.
>
> (2) Cryptography-based technologies (such as MPC) could support not only the similarity computations required by semantic retrieval, but also the exact term matching essential for lexical retrieval, addressing the key limitations of DP-based approaches.
> This is because cryptography-based technologies do not alter the raw data, while DP involves perturbing the data with irreversible noise, which makes it difficult to perform exact matching.
>
>
> [1] Shenglai Zeng, Jiankun Zhang, Pengfei He, Yue Xing, Yiding Liu, Han Xu, Jie Ren, Shuaiqiang Wang, Dawei Yin, Yi Chang, et al. The good and the bad: Exploring privacy issues in retrieval-augmented generation (rag). arXiv preprint arXiv:2402.16893, 2024.
>
> [2] Yangsibo Huang, Samyak Gupta, Zexuan Zhong, Kai Li, and Danqi Chen. Privacy implications of retrieval-based language models. arXiv preprint arXiv:2305.14888, 2023.
>
>
>
> *  *Weakness 3. Strict Privacy Definition.*
>
> **Response to Weakness 3.**
> This is a valid and important question.
> We add a new Section (Section 3 in our revised paper) that defines our privacy setting. We mainly consider a semi-honest setting where the privacy of both the query and documents is protected during the retrieval process in the RAG system. Our proposed framework can be integrated with various secure LLM inference frameworks to provide end-to-end privacy from query to response in the whole RAG system.

---

### Official Review · Reviewer_J7a5 · 2025-11-02

**Soundness:** 2
**Presentation:** 2
**Contribution:** 2
**Rating:** 4
**Confidence:** 2

**Summary:**

This paper presents Pisces, a framework for cryptography-based privacy-preserving retrieval-augmented generation (RAG). The goal is to protect both user queries and knowledge base documents while maintaining retrieval efficiency and accuracy. Pisces integrates dual-path retrieval — combining semantic and lexical matching — with novel cryptographic mechanisms. The semantic path adopts a coarse-to-fine strategy using an oblivious filter over Hamming distance and secure multi-party computation (MPC) for cosine similarity, while the lexical path employs a multi-instance labeled private set intersection (PSI) protocol to compute BM25 scores privately. The authors claim that Pisces achieves retrieval accuracy close to plaintext baselines (within 1.14%) and substantially improves efficiency (up to 496× faster in certain PSI tasks).

**Strengths:**

1. Comprehensive framework design: The paper systematically integrates multiple cryptographic primitives (MPC, PSI, OPRF, OKVS) into a cohesive RAG retrieval pipeline, ensuring privacy for both queries and documents.

2. Dual-path retrieval consideration: Supporting both semantic and lexical retrieval paths under privacy-preserving constraints is an ambitious and meaningful direction for improving private RAG systems.

**Weaknesses:**

1. Limited novelty and unclear challenge motivation: The paper’s technical contributions appear incremental, and it lacks a clear explanation of the key challenges being solved. The Introduction mostly summarizes system components and prior cryptographic methods, but does not articulate the specific research problem or insight that distinguishes Pisces from existing privacy-preserving RAG frameworks. Overall, it reads more like a technical report than a research paper with a well-defined innovation.

2. Unclear use of “cryptography-based” in the title: Although the framework is described as cryptography-based, the cryptographic novelty is not obvious. From the description, the “cryptography” aspect mainly involves using an embedding model and tokenizer to encode queries and documents, which is standard in most RAG architectures. If the cryptographic contribution only lies in combining existing MPC/PSI primitives, it does not sufficiently justify the “cryptography-based” claim. The authors should clarify how Pisces provides new cryptographic advances beyond typical embedding-based privacy mechanisms.

3. Missing experimental comparison with Table 1 baselines: The paper introduces several baseline methods in Table 1 (e.g., DP-RAG, LPRAG, RemoteRAG, etc.), but these methods are not actually compared in the experimental section. Without quantitative results against those baselines, it is difficult to assess whether Pisces offers meaningful advantages in accuracy, efficiency, or privacy protection.

**Questions:**

see the weakness

---

> ### Author Response · Authors · 2025-11-25
> **Response to Reviewer J7a5 [1/2]**
>
> Dear Reviewer J7a5,
>
> We sincerely thank the reviewer for your time and insightful feedback. We are grateful for the positive assessment of our work and the constructive comments, which help us significantly improve our paper. **All comments have been carefully addressed, and the corresponding revisions have been updated in the revised paper with major changes highlighted in purple for clarity**. We hope our revisions address all the concerns and bring the paper up to acceptance.
>
>
> Below, we provide point-by-point responses to each comment, along with additional explanations and experimental results where necessary.
>
>
> *  *Weakness 1. Limited novelty and unclear challenge motivation*
>
> **Response to Weakness 1.**
> Thank you for identifying clarity issues in our paper and constructive comments. To clearly demonstrate our novelty and challenge motivation, we have rewritten Section 1 (Introduction) in our revised paper. Below, we address your concerns point by point.
>
>
> 1. **Research Problem and Insights.** The research problem that we wish to answer is "How can we support dual-path retrieval while ensuring privacy for both the query and documents during the retrieval process?". To resolve this problem, we proposed Pisces, the first cryptography-based RAG retrieval framework that supports dual-path (semantic + lexical) retrieval while providing privacy protection for both the user query and documents in the knowledge base. The key insight distinguishing Pisces from existing DP-based privacy-preserving RAG frameworks is its ability to support lexical retrieval, which is difficult for DP-based frameworks due to irreversible noise addition that disrupts exact term matching. By leveraging cryptography-based techniques, which preserve data utility, Pisces maintains high retrieval performance while ensuring privacy.
>
>
> 2. **Clear Articulation of Research Challenges.** We explicitly identify and address two fundamental efficiency challenges when directly deploying existing cryptographic technologies approaches in the dual-path retrieval process:
> 	* **Challenge 1.** For the semantic retrieval, direct computation of similarities between a query and all documents in a large-scale knowledge base incurs prohibitive computation and communication overhead.
> 	* **Challenge 2.** For the lexical retrieval, state-of-the-art labeled private set intersection (PSI) methods require multiple invocations to obtain all necessary term frequencies for BM25 scoring, leading to significant computational overhead.
>
> 3. **Technical Contributions & Novelty.** To address these two challenges, we introduce two novel, customized cryptographic protocols for significant efficiency improvements.
> 	* **Technical Contribution 1.** For the semantic retrieval, we adopt a coarse-to-fine strategy with a core novel oblivious filter protocol that privately selects a candidate set to substantially reduce the search space followed by a standard MPC-based cosine similarity computation on the candidate set.
> 	* **Technical Contribution 2.** For the lexical retrieval path, we design a multi-instance labeled PSI protocol that obtains all necessary term frequencies in a single execution, avoiding the overhead of repeated labeled PSI invocations.
>
> *  *Weakness 2. Unclear use of “cryptography-based” in the title*
>
> **Response to Weakness 2.** The use of "cryptography-based" refers to a series of cryptographic protocols that protect both the user query and documents within a standard dual-path RAG architecture. As detailed in our response (Technical Contributions & Novelty) to Weakness 1, we introduce two novel cryptographic protocols, tailored to the requirements of semantic and lexical retrieval paths rather than simply combining existing MPC/PSI primitives.
>
> For the semantic retrieval path, we propose a coarse-to-fine strategy using a novel oblivious filter protocol that first privately selects a candidate set to substantially reduce the search space, followed by a standard MPC-based cosine similarity computation on the candidate set.
>
> For the lexical retrieval path, we design a multi-instance labeled PSI protocol that obtains all necessary term frequencies in a single execution, avoiding the overhead of repeated labeled PSI invocations.

---

> ### Author Response · Authors · 2025-11-25
> **Response to Reviewer J7a5 [2/2]**
>
> *  *Weakness 3. Missing experimental comparison with Table 1 baselines*
>
> **Response to Weakness 3.**
> This is a valid and important question. We have added experimental comparisons (Table 8 in Appendix B.7) against DP-based works in our revised paper. Experimental results show that our accuracy is significantly higher than that of DP-based works. As for RemoteRAG listed in Table 1, we will complete the experimental results and update them as soon as possible (before 03/12/2025).  Below, we show part of the experimental results and please refer to the revised paper for the complete results.
>
> Table 8 Retrieval accuracy comparisons between Pisces and baselines over ground-truth.
>
> | Dataset            |          | Framework           | Top-5 Semantic | Top-5 Lexical | Top-5 Dual-path | Top-10 Semantic | Top-10 Lexical | Top-10 Dual-path |
> |--------------------|---------------|---------------------|----------------|---------------|-----------------|-----------------|----------------|------------------|
> | **ClapNQ**         | Dev           | Plaintext           | 36.38%         | 47.42%        | 59.66%          | 42.47%          | 54.84%         | 67.30%           |
> |                    | Dev           | Pisces              | 30.14%         | 47.15%        | 58.05%          | 36.56%          | 54.35%         | 65.43%           |
> |                    | Dev           | DP-RAG              | 26.37%         | /             | 26.37%          | 36.92%          | /              | 36.92%           |
> |                    | Dev           | RandomProjection    | 15.45%         | /             | 15.45%          | 20.63%          | /              | 20.63%           |
> |                    | Train         | Plaintext           | 22.99%         | 36.78%        | 45.78%          | 27.69%          | 43.16%         | 52.10%           |
> |                    | Train         | Pisces              | 19.22%         | 36.72%        | 44.41%          | 21.95%          | 42.96%         | 50.75%           |
> |                    | Train         | DP-RAG              | 17.47%         | /             | 17.47%          | 23.79%          | /              | 23.79%           |
> |                    | Train         | RandomProjection    | 6.68%          | /             | 6.68%           | 9.04%           | /              | 9.04%            |
> |                    | Train_single  | Plaintext           | 27.57%         | 40.12%        | 50.56%          | 31.60%          | 46.92%         | 56.66%           |
> |                    | Train_single  | Pisces              | 23.44%         | 39.82%        | 49.39%          | 27.63%          | 46.83%         | 55.92%           |
> |                    | Train_single  | DP-RAG              | 21.31%         | /             | 21.31%          | 27.94%          | /              | 27.94%           |
> |                    | Train_single  | RandomProjection    | 4.41%          | /             | 4.41%           | 6.05%           | /              | 6.05%            |
> | **SQuAD**          | Dev           | Plaintext           | 33.80%         | 91.90%        | 93.30%          | 40.70%          | 94.60%         | 95.70%           |
> |                    | Dev           | Pisces              | 25.20%         | 91.60%        | 93.10%          | 31.80%          | 94.40%         | 95.50%           |
> |                    | Dev           | DP-RAG              | 24.74%         | /             | 24.74%          | 34.86%          | /              | 34.86%           |
> |                    | Dev           | RandomProjection    | 14.78%         | /             | 14.78%          | 19.94%          | /              | 19.94%           |
> |                    | Train         | Plaintext           | 26.40%         | 81.10%        | 84.80%          | 34.80%          | 85.10%         | 89.00%           |
> |                    | Train         | Pisces              | 17.40%         | 80.90%        | 84.10%          | 22.10%          | 85.00%         | 88.40%           |
> |                    | Train         | DP-RAG              | 20.34%         | /             | 20.34%          | 30.12%          | /              | 30.12%           |
> |                    | Train         | RandomProjection    | 9.08%          | /             | 9.08%           | 12.20%          | /              | 12.20%           |

---

### Author Response · Authors · 2025-12-03
**Summary**

Dear AC,

We sincerely thank you and the reviewers for your time and insightful comments. We are particularly grateful to Reviewer Nqrz (confidence 5) for increasing his/her rating on November 25. All reviewers' comments have been carefully addressed, and the corresponding revisions have been updated in the revised paper, with major changes highlighted in purple for clarity.
Below, we provide a summary of our key revisions.

1. **Novelty & Motivation (Reviewer J7a5, 24E7, Nqrz)** To better demonstrate our novelty and motivation, we have rewritten Section 1 (Introduction) in our revised paper. Our revisions have adequately addressed the main concerns of the three reviewers (J7a5, 24E7, Nqrz), as Reviewer Nqrz increased his/her rating based on our discussions. Specifically, we have

    * **Updated the motivating example to highlight the privacy concerns in RAG systems.** We consider the motivating example of personalized healthcare diagnostics to demonstrate the necessity of protecting the privacy of both the user query and documents in the knowledge base.

    * **Clarified the difference between cryptography-based and DP-based approaches.**  Cryptography-based approaches (such as MPC) could support both the similarity computations required by semantic retrieval and the exact term matching essential for lexical retrieval, while DP-based approaches involve perturbing the data with irreversible noise, making exact matching difficult.

    * **Explicitly identified two fundamental efficiency challenges when directly deploying existing cryptographic approaches to the dual-path retrieval process.** Large computation and communication overhead for similarity computations in semantic retrieval and multiple labeled PSI invocations for BM25 scoring in lexical retrieval.

    * **Detailed introduction of our proposed two novel, customized cryptographic protocols to resolve the two fundamental efficiency challenges.** An oblivious filter that privately selects a candidate set in semantic retrieval, and a multi-instance labeled PSI that retrieves all required term frequencies in a single execution in lexical retrieval.

2. **Experimental Comparison with Table 1 Baselines (Reviewer J7a5， Nqrz)** We have added experimental comparisons against DP-based approaches in our revised paper (see Table 8 and Table 9 in Appendix F). Experimental results demonstrate that our proposed framework achieves significantly higher retrieval accuracy compared to existing DP-based works. Reviewer Nqrz accepted these experiments and increased his/her rating.


3. **Strict Privacy Definition (Reviewer 24E7)** We have added a new Section, Section 3 (Problem Definition and Threat Model), in our revised paper that describes our privacy definition.

4. **Additional Large-Scale Experiments  (Reviewer EtQD)** We have conducted additional experiments using a large-scale dataset (hotpotqa_training with 1,795,146 chunks) and then updated Tables 2, 3, 8, 9, and Figures 2, 5 in our revised paper. These experiments confirm that our framework maintains high retrieval accuracy even at a large scale.


We thank you for your time and effort in reviewing our paper again. We believe our revisions have thoroughly addressed all concerns raised by the reviewers and brought the paper up to an acceptable level.

---

### Meta-Review · Area_Chair_MbUY · 2026-01-03

**Summary:**

This paper focuses on privacy preservation within the RAG system. Specifically, the authors develop cryptography-based approaches, such as MPC, to achieve this goal. While the paper provides theoretically guaranteed privacy protection, there is still room for its improvement. For instance, the reviewers have raised several issues associated with this work, such as unclear motivation, lack of contribution, missing baselines, and limited implementation. During the rebuttal, the authors have addressed most of these concerns and improved this paper. Given this, I tend accept this paper.

**Reviewer Concerns:**

During the rebuttal process, the reviewers raised several questions regarding this work. The authors have addressed the issues related to unclear motivation, implementation scope, and baseline comparison. There is no obvious drawback to this work.

**Reviewer Scores:**

I think Reviewer Nqrz might improve his score from 4  to 6, as the authors have adequately addressed the concerns. And others may maintain their original scores.

---

### Decision · Program_Chairs · 2026-01-26

Accept (Poster)